# THE CELL MUST GO ON: AGAR.IO FOR CONTINUAL REINFORCEMENT LEARNING

## ABSTRACT

Continual reinforcement learning (RL) concerns agents that are expected to learn continually, rather than converge to a policy that is then fixed for evaluation. Such an approach is well suited to environments the agent perceives as *changing*, which renders any static policy ineffective over time. The few simulators explicitly designed for empirical research in continual RL are often limited in scope or complexity, and it is now common for researchers to modify episodic RL environments by artificially incorporating abrupt task changes during interaction. In this paper, we introduce `AgarCL`, a research platform for continual RL that allows for a progression of increasingly sophisticated behaviour. `AgarCL` is based on the game Agar.io, a non-episodic, high-dimensional problem featuring stochastic, ever-evolving dynamics, continuous actions, and partial observability. Additionally, we provide benchmark results reporting the performance of DQN, PPO, and SAC in both the primary, challenging continual RL problem, and across a suite of smaller tasks within `AgarCL`, each of which isolates aspects of the full environment and allow us to characterize the challenges posed by different aspects of the game. Similarly, we evaluate three continual learning methods, Shrink and Perturb, ReDo, and Continual Backpropagation, and observe that they seem to provide a slight advantage over the traditional RL algorithms in AGARCL.

## 1 INTRODUCTION

Continual reinforcement learning (RL) is the RL setting where the agent is expected to learn continually, rather than the more traditional setting where the agent learns a policy that is then fixed for evaluation or deployment. Continual RL can be seen either as a problem formulation (Khetarpal et al., 2022; Abel et al., 2023; Kumar et al., 2025) or as a solution method for problems perceived by the agent as non-stationary. Such problems are often motivated by the big world hypothesis, which states that the "world" is bigger than the agent (Javed & Sutton, 2024); in this case, continual adaptation is simply a more effective approach than any fixed policy (Sutton et al., 2007; Janjua et al., 2024).

Much of the progress in RL has been driven by empirical advances, with experimental results often shaping algorithmic innovations and research directions. As a result, evaluation platforms play a central role in accelerating progress in the field (e.g., Todorov et al., 2012; Bellemare et al., 2013; Beattie et al., 2016). In continual RL, most evaluation platforms are adaptations of environments from traditional RL research, modified to reflect the idea that the world is bigger than the agent. This is typically achieved by artificially introducing some non-stationarity to the environment—such as periodically switching the problem faced by the agent (Powers et al., 2022; Abbas et al., 2023; Anand & Precup, 2023; Tomilin et al., 2023)—or, more rarely, by designing new environments tailored specifically to continual RL research, with no vestigial notion of episodes (Platanios et al., 2020).

However, both approaches have limitations. While reusing existing environments—especially complex simulators—is appealing, introducing non-stationarity through hand-crafted switches that are disconnected from the agent's behaviour or the environment's structure can feel artificial. Although we can sometimes experience abrupt phase shifts in our everyday life, smooth, gradual changes are at least as common. Nevertheless, this setting without abrupt changes is not represented in the available environments for continual RL research. The second—designing environments specifically for continual RL—is promising in spirit, but existing instantiations tend to be limited in complexity or scope. Our work takes a complementary direction, aiming to preserve these purpose-built environments' continual, non-episodic nature while introducing richer, ever-evolving

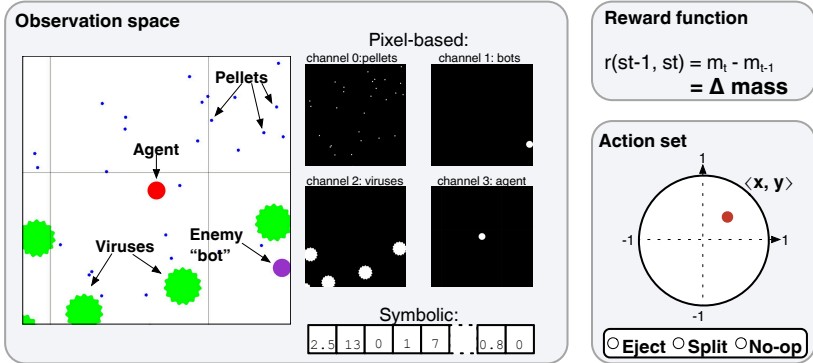

Figure 1: Agent-environment interface and main entities in `AgarCL`. The agent has access to one of two observation types: pixel-based or symbolic. The pixel-based observation includes four channels that represent different game entities: pellets, bots, viruses, and the agent itself. The agent can receive a symbolic observation, consisting of pre-processed features such as the distances to nearby enemies, pellets, and other entities. The reward function is defined as the change in the agent's mass between two consecutive time steps. The action space in `AgarCL` is hybrid: at each time step, the agent selects an $\langle x, y \rangle$ coordinate mimicking where a human player would point their mouse. Moreover, the agent decides whether to split, eject mass, or just move. In Section 3, we provide detailed descriptions of these entities and the game dynamics, including the available actions.

stochastic dynamics, continuous actions, partial observability, resource-driven competition, and a scaffolded progression of behavioural complexity in a spatially structured world.

Specifically, in this paper, we introduce a new evaluation platform based on the game Agar.io (see Figure 1). This platform, called `AgarCL`, was developed with a heavy emphasis on supporting research on continual RL. In this environment, the agent controls circular cells within a bounded Petri dish-like arena. The agent perceives the world through high-dimensional, pixel-based observations,[1] and acts through both continuous and discrete actions. Navigation is determined by continuous actions represented as $\langle x, y \rangle$ coordinates, while additional affordances such as *splitting* or *ejecting mass* are defined as discrete actions. In `AgarCL`, the reward is the difference in the agent's mass between two consecutive time steps. Thus, an effective agent has a forever-growing mass.

The dynamics of the game Agar.io are what make this environment interesting from a continual RL perspective. The agent "lives" in an environment full of other agents.[2] Agents can increase their mass by either collecting stationary food pellets or absorbing other agents that are smaller than they are. Consequently, an agent's primary objective is not only to increase its mass, but also to avoid larger cells that pose a threat. At each time step, the mass of a cell naturally decays, with the rate of decay increasing with size—because agents have the same density in the game, more mass implies greater size. Aside from its potential never-ending nature, and the complexity induced by other agents' behaviours, the key property of `AgarCL` that makes it interesting from a continual RL perspective is how its dynamics change according to the agent's mass. Larger agents move more slowly. Moreover, the observation space changes as the agent increases mass—it zooms out, covering a larger area to keep the agent's body visible. The consequence of an agent's action (and even its observation stream) is constantly changing in a relatively smooth manner. Abrupt changes are also present—such as when the agent absorbs another agent, drastically increasing its mass; when it splits itself into multiple cells on purpose; or when it splits due to a virus. Figure 2 and a Youtube video[3] depict some of the environment dynamics supported by `AgarCL`.

We demonstrate the feasibility of `AgarCL` as an evaluation platform and a challenge problem for continual RL by benchmarking three widely used algorithms, DQN (Mnih et al., 2015), PPO (Schulman et al., 2017), and SAC (Haarnoja et al., 2018), across multiple variations of the environment.

---

[1] Alternative observation types, with a more symbolic flavour, providing backwards compatibility to other platforms (Zhang et al., 2023), are also supported. We further discuss this option in Section 3.

[2] In `AgarCL`, the behaviour of other agents is determined by hand-coded policies. Future work will focus on supporting learning across multiple agents, making this environment multi-agent.

[3] https://www.youtube.com/watch?v=CGpvzHIqFLA

We also benchmark three approaches directly tailored to continual learning, Shrink and Perturb (Ash & Adams, 2020), ReDo (Sokar et al., 2023), and Continual Backpropagation (Dohare et al., 2024). In the core game, we find that none of these algorithms are able to learn an effective policy. We also evaluate the performance of the traditional RL approaches on several mini-games within `AgarCL` that isolate different aspects of the environment's complexity, such as non-stationarity, exploration, and credit assignment. These experiments provide indirect insight into the sources of failure in the full game. Additionally, we present empirical findings that highlight persistent challenges in continual RL research, particularly in evaluation methodology and sensitivity to hyperparameters.

## 2 BACKGROUND

We use the RL formalism to describe the sequential decision-making problem posed by `AgarCL`. In this problem, interactions take place in discrete time steps. The agent starts in a state $S_0 \sim \mu$, where $\mu$ is a start-state distribution, and $S_0 \in \mathcal{S}$. However, the agent only has access to an observation, $\omega_0 \in \Omega$, generated by an observation function, $\Phi : \mathcal{S} \to \Delta(\Omega)$. At each time step, $t$, the agent takes an action, $A_t \in \mathcal{A}$, and receives an observation, $\omega_{t+1} \in \Omega$, and a reward signal, $R_{t+1} \in \mathbb{R}$. Actions are chosen according to the agent's policy, $\pi$, a (possibly stochastic) function of previous observations. The agent's goal is to maximize some variant of the expected return, $G_t$, that is, of the expected total sum of rewards it receives. The most common variant is the *discounted* return, $G_t^\gamma$, in which later rewards are discounted by $0 \leq \gamma < 1$, such that $G_t^\gamma \doteq \sum_{t=0}^\infty \gamma^k R_{t+k+1}$. Alternatively, one may consider the *average-reward* setting: $G_t^{\bar{r}} \doteq \sum_{k=0}^\infty \left( R_{t+k+1} - r(\pi) \right)$, where $r(\pi) \doteq \lim_{h \to \infty} \frac{1}{h} \sum_{t=1}^h \mathbb{E}[R_t | \omega_0, A_{0:t-1} \sim \pi]$, with $\omega_0$ denoting the initial observation.

The environment evolves according to the transition function $p : \mathcal{S} \times \mathcal{A} \to \Delta(\mathcal{S})$, which determines how the state changes at each time step given an action. In Markov Decision Processes (MDPs; Puterman, 2014), there is no distinction between observations and states; the observation function can be considered the identity. However, as we discuss below, `AgarCL` is a partially observable environment, with complex dynamics hidden from the agent, and is best described using the full POMDP formalism (Kaelbling et al., 1998). This partial observability causes the agent to perceive the environment as changing, whether due to a potential source of non-stationarity in the transition dynamics, or simply due to partial observability. As mentioned above, our underlying assumption is that continual RL is beneficial in this setting because it will be more effective than than any fixed policy (see Sutton et al., 2007; Janjua et al., 2024). This also allows us to avoid placing restrictions on agents just to ensure the environment is "bigger than the agent" (Javed & Sutton, 2024).

In this work, we use the undiscounted return as the performance metric, but we evaluate algorithms designed for the discounted formulation, even though average reward is arguably better suited to continuing tasks. Because our focus is on `AgarCL` as a research environment, not on solution methods, we decided to leverage the much more mature literature on deep RL in the discounted case. Note, this is not uncommon. In Atari 2600 games, for example, agents are evaluated on the score they accumulate while they are trained to maximize the discounted expected return (Bellemare et al., 2013; Machado et al., 2018). Additionally, for every problem, there exists a critical discount factor such that any solution using a higher value will maximize the average reward (Blackwell, 1962).

## 3 AGARCL: AGAR.IO FOR CONTINUAL REINFORCEMENT LEARNING

We begin by discussing our main contribution: Agar.io for Continual RL (`AgarCL`). This is an environment for research in sequential decision-making that supports settings in which the environment is constantly changing, but not necessarily too abruptly. This setting can potentially instantiate the big world hypothesis (Javed & Sutton, 2024), and is a setting in which learning continually is a more effective solution than a fixed policy learned for a predetermined number of steps.

### 3.1 THE GAME: AGAR.IO

Agar.CL is based on the multiplayer online game Agar.io in which each player controls one or more circular cells. The game draws an analogy to a Petri dish containing interacting cells, food sources, and viruses. Players aim to grow by absorbing smaller entities, such as static pellets and other players' cells, while avoiding larger opponents and strategically interacting with viruses, which can either fragment or shield cells depending on their size (see Figure 1). Players can also split their cells

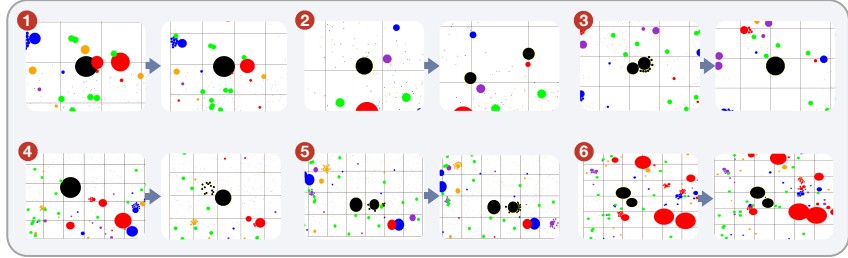

Figure 2: Environment dynamics and actions. ① The agent (in black) can eat smaller cells to gain mass. ② The SPLIT action divides each of the agent's cells in half and propels them in a chosen ⟨x, y⟩ direction, allowing slower agents to catch faster ones. Each cell moves at a speed inversely related to its mass. ③ Cells can later merge if brought close together. Depending on its mass, the agent can either ④ be split by a virus or ⑤ consume it. ⑥ The agent can also EJECT mass in a chosen direction. This mass can be consumed by any cell, including the agent. Ejecting enough mass into a virus spawns a new virus and propels the original, enabling smaller cells to attack larger ones.

for tactical reasons, enabling simultaneous control of multiple cells. The game's rules are simple, but complex dynamics emerge from these interactions. Figure 2 depicts some of these game dynamics.

The arena has three types of entities: *pellets*, *cells*, and *viruses*. Pellets are randomly scattered, static, have a fixed size, and grant 1 mass when consumed. Cells (or bots), can consume pellets, viruses, and smaller cells. When consuming another cell, the player gains its mass. Viruses have a mass of 100; depending on the player's size, a virus can either be absorbed by the cell or cause it to split.

Players start as a small cell of mass 25. If the player's cell(s) is consumed by a larger cell, the player's cell is eliminated from the game. The player then respawns with the same initial mass. While this could be perceived as an episodic task due to the repeated nature of the interaction, more successful agents are expected to "live" much longer and not see such resets. Additionally, the cell that absorbed the player's cell maintains its new mass after the player has respawned, such that actions from a previous "episode" impact the new one. Currently, our implementation supports the single-player setting; the other players, called *bots*, follow heuristic-based behaviour. We made this choice to focus on the continual aspect of the problem. We refer the reader to the GOBIGGER (Zhang et al., 2023) research platform if they are interested in the multi-agent aspect of the game.

Many sources contribute to the ever-changing nature of the game. New pellets are generated at every 600 ticks (10 seconds) while fewer than 500 pellets are in the arena. Likewise, new viruses are generated at every time step whenever the total virus count is below 10. Additionally, every cell belonging to an agent loses 0.2% of its mass each second. As cells grow, they move more slowly. Their speed, $v$, is determined by the function $v = \text{mass}^{0.439}$. Thus, smaller players must consistently evade larger players, steer clear of corners, or use viruses for defense. Players can feed viruses until they are large enough in order to split them toward larger players. Typically, a split occurs if a player feeds a virus seven times. The player's field of view also varies according to its mass, as the game needs to depict all of its cells.[4] Figure 3 depicts an example of how an Agar.io game can progress.

Due to all these dynamics, players must continually balance the need for immediate gains with potential long-term risks. A common tactic involves splitting a large cell to increase the chances of consuming smaller opponents, who might otherwise be out of reach due to the large cell's slower speed. However, this aggressive move introduces significant risks: the newly split, smaller cells can be quickly eaten by larger opponents or even by other small, agile players. This delicate balance between offense and defense, combined with the need to manage mass effectively, forms the core of competitive play. The game's design inherently promotes these strategies by ensuring that no single approach guarantees success in every situation, requiring the agent to continually adapt to survive.

---

[4]There are many additional rules in the game. When an agent has split into 14 cells, it can no longer split into more. To prevent larger players from simply consuming all viruses, a penalty is activated when an agent consumes 3 or more viruses in a row within a minute. After hitting 3 viruses, the rate of mass decay increases, accelerating further with each additional virus consumed. This penalty persists across respawns. Some are likely unnecessary to capture the spirit of the game, but we mention them to emphasize our focus on replicating the original game, which is interesting to people and free of experimenter bias as it was not designed for AI research.

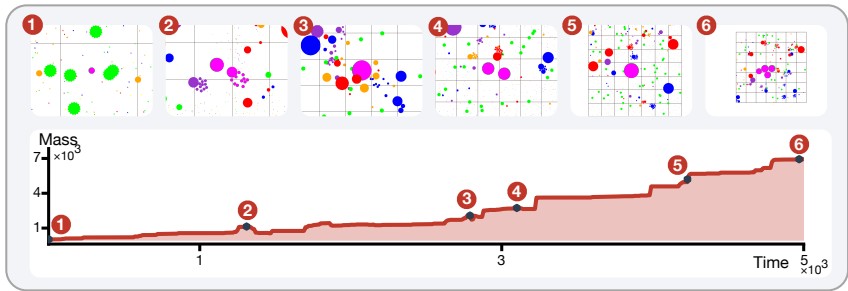

Figure 3: The first 5,000 time steps of an expert trajectory showing game progression. The agent (in pink) steadily gains mass: ① starting small, it eventually passes over a virus; ② splits into smaller cells; ③ grows large enough to consume viruses and other agents; ④ splits to attack, increasing its speed. As it grows, the view zooms out to show its full body. ⑤ The agent ultimately surpasses all opponents in mass and ⑥ can see the entire arena. Run recorded while one of the authors played.

### 3.2 AGENT-ENVIRONMENT INTERFACE

Once the environment dynamics are defined, we must describe how an agent can interact with such an environment. We follow the standard RL formalism where the agent-environment interaction takes place in terms of actions, observations, and rewards. We describe those below and in Figure 1.

**Reward Function.** To keep the reward bounded, we define the reward function as the difference in the agent's mass between two consecutive time steps. Formally, this is expressed as:

$$R_t = m_t - m_{t-1}. \tag{1}$$

Importantly, the problem is continuing (non-episodic), so the agent's death does not terminate the game. Upon respawning, the agent receives a reward equal to its death mass less its initial mass. This choice discourages death as a cost-free reset when the agent is bigger than its initial mass.

**Observation Space.** How the agent perceives the environment is key in any RL problem. Here, we emphasize a pixel-based top-down view of the environment. Such a choice requires the agent to deal with a high-dimensional, partially observable and ever-changing observation stream. The figures throughout the paper provide many examples of renderings of the actual observations received by the agent, with Figure 3 particularly emphasizing the evolution of such an observation stream.

Importantly, variants of Agar.io have been used in RL research before (Wiehe et al., 2018; Ansó et al., 2019; Zhang et al., 2023), but the setting with high-dimensional pixel-based observations has never been studied (or supported). In previous work, the observation always consisted of some variation of a grid-like observation. Although we do not provide results in such a setting, AgarCL also supports using more symbolic representations such as these. We discuss both choices below.

*Pixel-Based Observation:* We represent the game screen, $\mathbf{O}_t$, as a tensor $\mathbf{O}_t \in \mathbb{R}^{N \times N \times 4}$, where $N \times N$ denotes the spatial resolution of the game screen (by default, $N = 128$). The third dimension corresponds to separate channels for pellets, viruses, enemies, and the agent (including gridlines). Fig. 1 depicts an example of such an observation (the gridlines are too faint to be seen).

*Grid-Like Observation:* We adapted the GoBigger observation (Zhang et al., 2023) to the single-agent setting. It is divided into two main parts: the global state and the player state. The global state captures information such as the map size, the number of frames in the game, and the number of frames that have passed. In the player state, we focus solely on the current agent's information, including its field of view, visible entities in that space, the agent's score, and its available actions. The overlap field is critical, as it captures details about nearby pellets, viruses, and cells (each with associated positions, velocities, and other necessary attributes).

**Action Space.** The action space is hybrid, and an agent can perform two types of actions simultaneously. The agent controls its cells by selecting a point on the screen, mimicking a human's cursor movement, which determines the direction in which all of its cells move. The range of these *continuous* actions is between $[-1, 1]$ in two dimensions, $\langle x, y \rangle$. Simultaneously, the agent needs to make a *discrete* choice between splitting, ejecting pellets, or simply moving, with no further discrete action.

The split action divides a cell into two equal parts, each with half the original mass, provided the produced cells have at least a mass of 25 (otherwise, it has no effect). One of the newly split cells

is propelled toward the cursor with significant momentum. After splitting, the player must manage multiple cells simultaneously, using the cursor to navigate each cell. The eject action ejects a small mass (called a pellet) from each cell toward the cursor. They can be consumed by cells or viruses. Every action taken is repeated four times, and the observation received by the agent is generated by the environment after the execution of the selected action(s) four times; we call this value *frame skip*.

Finally, `AgarCL` has stochastic dynamics. Noise sampled from a normal distribution, $\mathcal{N}(0, 1)$, is added to the continuous actions the agent sends to the environment.

**Simulation Speed.** In our experiments, when looking at the interquantile mean (IQM) over ten independent trials, a random agent, written in Python, receives 2,016 frames per second with a frame skip of 1. With a frame skip of 4 (default), the agent receives 1,163 observations per second, which represents 4,652 frames in the game and 1,163 actions selected by the agent. Such a speed, orders of magnitude faster than benchmarks such as GoBigger, was achieved through a series of technical contributions ranging from the use of the EGL API for faster rendering, efficient implementations of collision detection, observation generation, and memory management.

**Technical Details.** Appendix C has details about `AgarCL`'s release, interface, and performance.

## 4 EXPERIMENTS AND RESULTS

To evaluate the learning capabilities of agents in `AgarCL`, we first use DQN (Mnih et al., 2015), PPO (Schulman et al., 2017), and SAC (Haarnoja et al., 2018).[5] These algorithms span both value-based and policy-gradient methods to represent the performance of different classes of algorithms.

Each agent has an encoder component which processes $128 \times 128 \times 4$ raw images. The encoder comprises three convolutional layers with kernel sizes of $8 \times 8$, $4 \times 4$, $3 \times 3$, and strides of 4, 2, and 1. Each convolutional operation is followed by layer normalization (Ba et al., 2016) and a rectified linear unit (ReLU). The final output, a $32 \times 12 \times 12$ feature map, is flattened and passed through a fully connected layer to produce a compact embedding. This embedding serves as the basis for predicting both discrete and continuous actions, supporting hybrid control in the `AgarCL` environment. In Appendix G, we discuss the adaptations we made to the algorithms we evaluated.

### 4.1 BENCHMARK RESULTS FOR CONTINUAL RL IN `AgarCL`

The full game is the reference setting and default setup we are introducing. It is a challenge problem that poses many difficulties, as we discuss in the next sections. The arena size is $350 \times 350$ and it contains ten viruses, eight bots, and 500 pellets, which are regenerated every 600 ticks. We evaluated DQN, PPO, and SAC learning over 160 million frames in the environment, averaging over 10 independent runs. Running SAC, for example, takes more than seven days in this setting.

The empirical evaluation we performed strongly supports our claim that `AgarCL` is a challenging evaluation platform for traditional RL algorithms, as they all fail to learn an effective policy in this environment. Figure 4 depicts the learning curves obtained by each algorithm. We provide exact numerical values in Table 10 in Appendix J.

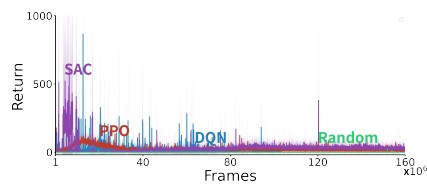

Hyperparameters are an essential aspect of every RL algorithm, and it is important to discuss hyperparameter tuning in continual RL problems. In truly continual RL, the tuning strategies commonly used in RL are impractical (Mesbahi et al., 2024). How can one tune the hyperparameters of an agent that is expected to live forever? The obvious alternative, tuning hyperparameters for a chosen shorter interval is problematic because we incur the risk

Figure 4: Performance of established deep RL algorithms in the full game of `AgarCL`. Each curve is the average over 10 seeds, with a moving average computed over a window of 100 steps.

of overfitting to that horizon, going against the idea of continual learning. This matters in practice; for example, the hyperparameters used for 100k frames in Atari differ greatly from those used by agents when training for 200M frames. Moreover, the performance of an algorithm is often hugely impacted by minor hyperparameter variations, a topic we further discuss in Appendix I.2.1.

---

[5]Our implementations are available in `https://github.com/AgarCL/AgarCL-Benchmark`.

Due to the difficulty of tuning hyperparameters over a potentially unbounded timeframe and to avoid overfitting to a specific horizon, we used the best hyperparameters we found for AgarCL's continual MINI-GAME 4, the setting most similar to this one (but much shorter), as discussed below.

## 4.2 AgarCL as a Continual Reinforcement Learning Testbed

We have not yet provided evidence to support our claims that AgarCL is a potentially promising evaluation platform for continual RL research. We do so here. Specifically, we demonstrate that fixing an agent's policy results in a decrease in performance compared to a continually learned policy.

It is hard to show that fixing the agent's policy leads to worse performance when the algorithms we evaluated failed to learn a reasonable policy. Thus, only to evaluate the impact of continual learning in AgarCL, we evaluated PPO in an environment in which pellets and viruses regenerate more frequently. Pellets and viruses are regenerated every 120 ticks instead of 600. We also varied the pellet density in the arena to obtain an easier setting for this analysis (1024 instead of 500). We evaluated PPO in this setting because it consistently outperformed the other baselines across various settings, as discussed below. Shortly, we observed that making more pellets available to the agent in the environment, along with the other changes mentioned above, led to learning curves that go above zero. The performance of these agents for different numbers of pellets is available in Appendix K.

Having found an easier setting where we can evaluate existing agents, we now evaluate the performance of a fixed policy learned by a PPO agent trained for 32 million and 48 million training steps. We evaluated these policies in the setting above with 1024 pellets in the arena. As shown in Figure 5, although these fixed-policy agents initially perform competitively, the performance of frozen-policy agents collapses after a certain point. This result further supports our claim that fixed policies might not be able to adapt to the non-stationary nature of AgarCL, and that it might be useful for continual RL research.

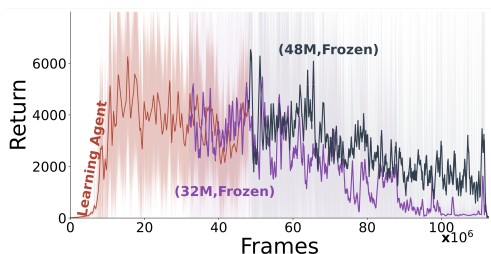

Figure 5: Performance of fixed-policy agents initialized from checkpoints at 32M and 48M steps. We report the moving average over 10 random seeds with a window size of 1000 steps.

It is also interesting to point out settings in which the drop did not happen. While looking for an easier setting, we did the same experiment with fewer bots in the arena (four instead of eight). In that case, we observed that PPO was able to learn a relatively stable policy that would collect some pellets and consistently avoid the other bots (see Appendix L). This further supports the importance of the different artifacts in the environment to obtain an evaluation framework for continual RL agents.

## 4.3 Existing Continual RL algorithms in AgarCL

Although the primary purpose of this paper is not to test, let alone benchmark, continual RL methods in AgarCL, one might wonder if existing solutions would naturally be effective here.

Most methods rely on the notion of tasks and are not directly applicable to AgarCL. Examples include EWC (Kirkpatrick et al., 2017), MAS (Aljundi et al., 2018), and LwF (Li & Hoiem, 2016). EWC and MAS use task information in the regularization constraints on parameter updates, while LwF relies on tasks to define distillation losses to preserve performance on previous tasks.

Thus, we focused on generic methods that we could apply to AGARCL without modification. Such methods tend to focus on maintaining plasticity during learning. Specifically, we augmented PPO with Shrink and Perturb (Ash & Adams, 2020), ReDo (Sokar et al., 2023), and Continual Backpropagation (Dohare et al., 2024). We considered the easier setting from Section 4.2.

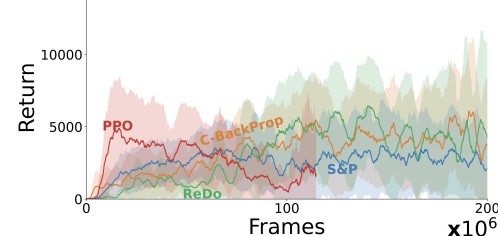

Figure 6: Performance of different cont. learning approaches. We report the moving average over 10 runs with a window size of 10,000 steps.

As shown in Figure 6, the three methods[6] perform similarly and despite the fact their performance is extremely noisy, they seem to exhibit a better performance trend than PPO. Details about the hyperparameters we used, as well as a table with the results depicted in Figure 6 are available in Appendix L. These results suggest that existing continual learning solutions might indeed be effective in AGARCL, although a much more careful analysis would be required to allow for definitive claims, something we consider to be outside the scope of our paper. Importantly, we do not expect any of these methods to suffice in AGARCL as they do not tackle key challenges posed by the environment, which we discuss below. In fact, our attempts to have these continual learning methods succeed in the benchmark setting (500 pellets regenerated every 600 ticks) all failed.

## 4.4 VALIDATING AGARCL THROUGH MINI-GAMES

As we established above, `AgarCL` is a very challenging environment. It not only requires agents to be able to reliably learn continually, which is already a significant challenge by itself, but also to address open problems in continual RL research, such as exploration without resets, long-term credit assignment, and representation learning in light of a varying observation stream.

To validate and understand the proposed environment, we take a step back and evaluate existing agents in a collection of mini-games we designed within Agar.io. These mini-games isolate specific challenges such as non-stationarity and the non-episodic nature of the problem. Although they can be used for diagnosing problems and understanding an agent's behaviour, we do not necessarily recommend that researchers use them as benchmark tasks instead of the default setup presented above.

Importantly, we tuned each algorithms' hyperparameters in every mini-game. We did so over 3 random seeds, selecting the configuration that maximized the mean return across the final 100 episodes of a 20-million-frame run. To avoid maximization bias, we evaluated this configuration with 10 new trials. Details on the hyperparameter ranges and selection criteria are discussed in Appendix H.

**Non-Stationarity through Mass Variation.** We first consider a set of progressively challenging mini-games that require the agent to collect pellets. The environment in such mini-games does not have other bots or viruses. For simplicity, this first set of mini-games is *episodic*. Episodes were 500 and 3000 time steps long (first three vs latter three mini-games) and had a single start state.

These mini-games were designed to evaluate the impact of three different aspects of the game: (i) (short-term) exploration, (ii) mass decay, and (iii) having a bigger mass, thus being much slower but able to split. To simplify exploration, delayed credit assignment, and partial observability, MINI-GAME 1 consists of collecting pellets laid out in a square path. The agent starts with a mass of 25, and there is no mass decay. This is the simplest mini-game, but it still features non-stationarity: when the agent gains mass, it becomes larger and correspondingly slower, but all the agent needs to learn is to follow the dense path of "breadcrumbs". MINI-GAME 2 is the same as MINI-GAME 1, but with mass decay (the negative amount received by the agent varies according to its mass); and MINI-GAME 3 is the same as MINI-GAME 2, but the agent's starting mass is 1000 (the agent is much slower, loses more mass per step, and can split). MINI-GAME 4, 5, and 6 match the first three mini-games, except that pellets are randomly scattered in the environment, not in a dense path the agent can blindly follow. In these latter three mini-games, credit assignment is more delayed, partial observability matters more, and exploration is slightly more difficult (although the resets upon episode termination still simplify the task). Screenshots of these mini-games are available in Appendix I.1.

Figure 7 presents the algorithms' performance across these mini-games. A table with numerical results is available in Appendix J. In the simpler problem, MINI-GAME 1, all approaches achieve a performance that is very similar to that of a human player. Interestingly, introducing the ever-changing dynamics already makes the problem much harder, even in the square-path task, as all agents achieve approximately half of the human performance here. Starting the agents with a larger mass makes the problem even more challenging. These insights carry over to the mini-games in which pellets are uniformly distributed throughout the arena. However, these mini-games seem to be a much harder setting for the algorithms we considered and, once mass decay is introduced, only PPO is able to learn *something*. Note that SAC is an algorithm 3–5$\times$ slower than PPO in terms

---

[6]There are other algorithms one could evaluate. One such example is CLEAR (Rolnick et al., 2019), which leverages past experience from the replay buffer to preserve plasticity and reduce catastrophic forgetting. Because we focus on the environment itself, we decided instead to evaluate the impact of different features of the environment, as we discuss in the next section, instead of exhaustively evaluating baselines in a single setting.

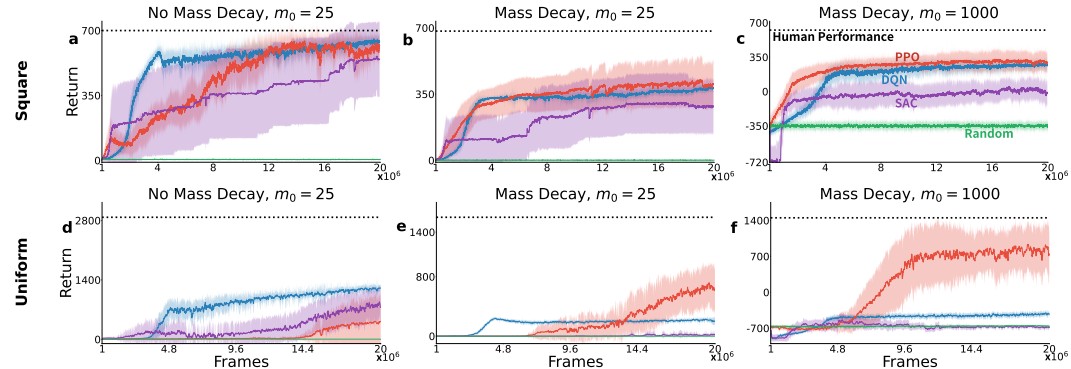

Figure 7: Performance of RL methods on *episodic* pellet-collection mini-games. Panels ⓐ, ⓑ, and ⓒ show the performance on *the square-path tasks* (mini-games 1, 2, and 3), while ⓓ, ⓔ, and ⓕ show the performance on *randomly regenerated tasks* (mini-games 4, 5, and 6). The y-axis scales vary across plots. The dashed line marks human performance, and the green line the random policy. The shaded region shows the 95% CI over 10 runs, computed using the t-distribution.

of wall clock time and, at least in our experiments, it is not more sample-efficient than PPO. [7]

**Continual Problems.** We now evaluate the mini-games discussed above in the continual case.[8] We removed the episodic resets, and we replenish all pellets every 600 frames. The maximum number of pellets is 500, which are replenished randomly (still within the path for the square-path mini-games). We re-tuned all hyperparameters in each mini-game for all algorithms evaluated.

These tasks turned out to be exceedingly difficult. Figure 9 and Table 10, with the performance of each algorithm, can be found in Appendix J. In the square-path tasks, no agent is able to succeed, partly due to exploration and partial observability. If the agent deviates from the square path and loses sight of it, the agent struggles to find the path back, and without resets, it must find the path back to continue learning. In this continual case, the uniform distribution of pellets makes the problem easier, maybe because the agent can always expect to see at least a few pellets. This is different when we reduce the number of pellets in the environment by half, a setting in which the baselines start to struggle. These results are available in Appendix K. Note that we trimmed the learning process in MINI-GAME 4 when the agents achieved the maximum allowed mass; otherwise, without mass decay, they would keep growing. As before, PPO is the most robust algorithm we considered.

Finally, it is natural to wonder if a change in network architecture to explicitly incorporate memory into the agent would not help tackle these problems. Thus, we also evaluated PPO augmented with a GRU (Cho et al., 2014) in these continual pellet-collection mini-games. However, we did not see any consistent improvement that would justify such an approach, and that is why we have focused on simpler architectures throughout this paper. We discuss these results in detail in Appendix I.2.2.

**Interacting with Other Agents.** We also considered mini-games where the agent is in the arena with another bot. We evaluated the agent against many different types of simple bots with a fixed policy. The agents we considered were never able to learn a policy that would collect enough pellets to outsize the other bot and then absorb it. This was true even for the bots that did not chase the agent. This is an example of how challenging exploration can be in such an environment, even for learning basic skills that any agent should have. We further discuss these results in Appendix I.3.

**Interacting with Viruses.** In addition to evaluating the agents' ability to collect pellets and to absorb other agents, we also designed a mini-game to evaluate whether an agent could learn to use viruses to split bigger bots and then absorb them. Shortly, as before, the answer is no. None of the agents was able to learn the sequence of actions required to succeed in such a task, even though we made it as easy as possible. These failures make it clear why related work would mostly focus on pellet-collection tasks in Agar-like environments. See Appendix I.4 for results and discussion.

---

[7]To obtain these results, we ran over 900 jobs lasting between 15 hours (DQN and PPO) and 23 hours (SAC).

[8]The word *continual* was deliberate. It emphasizes the non-stationary nature of the problem. *Continuing* often refers to stationary problems with an infinite horizon, and *continuous* problems imply continuous actions.

## 5 RELATED WORK

GOBIGGER (Zhang et al., 2023) is the approach closest to ours due to its Agar-style gameplay mechanics. However, GOBIGGER was designed to study collective behaviours in multi-agent RL. It focuses on supporting multiple *teams* of agents. It is not designed to support continual RL research; in fact, all of its tasks are episodic. Compared to GOBIGGER, `AgarCL` supports additional data streams and scales much better with more cells and pellets. Additionally, early work investigated the feasibility of using Agar-style games as RL environments, but with a heavy emphasis on evaluating some deep RL algorithms, mostly for pellet-eating tasks (Ansó et al., 2019; Wiehe et al., 2018).

Many other platforms have been used in continual RL research. `AgarCL` complements these, as it introduces features that are quite different from those of these platforms. A significant difference between `AgarCL` and most other platforms is the fact that it is not designed around the notion of *episodic tasks* that switch periodically like in Switching ALE (Abbas et al., 2023), Continual World (Wolczyk et al., 2021), POET (Wang et al., 2019), MEAL (Tomilin et al., 2025), Continual NavBench (Kobanda et al., 2025), and many MiniGrid problems (Chevalier-Boisvert et al., 2023).

The main platform supporting non-episodic continual RL research is JellyBean World (Platanios et al., 2020), but it has much simpler dynamics, a discrete action space, and minimal non-stationarity, primarily arising from abrupt changes in the reward function. Alternatives include bespoke adaptations of problems in MuJoCo (Todorov et al., 2012) and IsaacGym (Makoviychuk et al., 2021) used in specific investigations (e.g., Feng et al., 2022; Long et al., 2024) but that have never been made available as an evaluation platform. Irrespective of individual features, note that `AgarCL` integrates them in a way that is tightly coupled to the agent's state and behaviour, with emergent interactions between reward, perception, and dynamics. Replicating this effect with these physics engines would be non-trivial and would not naturally yield the same continual, smoothly evolving challenges.

Finally, one can view complex environments such as NetHack (Küttler et al., 2020) or Minecraft (Johnson et al., 2016; Guss et al., 2019) as possible evaluation frameworks for continual RL under the premise that such environments are "bigger than the agent" (Javed & Sutton, 2024). Approaching continual RL through these environments might make the problem unnecessarily harder. As interesting as Nethack is, it relies on language (and human knowledge), with existing approaches requiring additional machinery to succeed. Similarly, the vast majority of solutions in Minecraft rely on human demonstrations, given the difficulty of exploration in such an environment.

## 6 CONCLUSION

We introduced `AgarCL` as an evaluation platform for continual RL. `AgarCL` captures key challenges, such as partial observability and non-stationarity, while avoiding the abrupt task switches common in existing benchmarks. We evaluated DQN, PPO, and SAC in mini-games and the full game, demonstrating the challenges of the latter and how the former can support development and understanding. `AgarCL` has features that are important to advance research in continual RL, such as non-episodic interaction, smooth endogenous non-stationarity, high-dimensional observations, continuous actions, and potentially infinite horizon. It highlights the limitations of standard deep RL algorithms and the unique challenges of continual RL. Aside from introducing features mostly absent in other frameworks, we have shown that `AgarCL` poses significant challenges to traditional RL solutions, and that fixed policies seem to be unable to maintain stable performance in the environment.

This work has focused on the problem, not on existing solution methods. Thus, due to both scope and space considerations, we have refrained from focusing too much on algorithms introduced to tackle continual RL problems. A limitation of our work is the large amount of resources it required from us in the context of deep RL. Continual RL is particularly challenging given the timescale on which we want to evaluate our agents. Although one of our main concerns was to provide a fast simulator to the community, the experimentation cycle can still end up being quite long in `AgarCL`. In the Appendix A and E, we discuss the broader implications of our work and the computational resources we used.

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

## A  BROADER IMPLICATIONS

This work introduces a new evaluation framework for continual RL, built around the video game Agar.io. Given that this project is based on a video game and designed for benchmarking and evaluation purposes, there are no foreseeable negative societal impacts. As with most research involving simulated environments in AI, the risks are minimal and do not require significant mitigation.

## B  USE OF LARGE LANGUAGE MODELS

LLMs were only used for minor language polishing during the writing of the paper.

## C  SOFTWARE RELEASE AND TECHNICAL DETAILS

`AgarCL` is released as free, open-source software under the terms of the MIT license. It is implemented on top of the AgarLE (Deaton, 2018), an incomplete implementation of Agar.io with an OpenAI Gym (Brockman et al., 2016) interface. `AgarCL`'s source code is publicly available in:

https://github.com/AgarCL/AgarCL

`AgarCL`'s core simulation engine and is implemented in C++, rendering is handled by OpenGL. A Python interface, built with Pybind11, is also supported. In our experiments, when looking at the interquantile mean (IQM) over ten independent trials, a random agent, written in Python, receives 2,016 frames per second with a frame skip of 1. With a frame skip of 4 (default), the agent receives 1,163 observations per second, which represents 4,652 frames in the game and 1,163 actions selected by the agent. We report the IQM to discard fluctuations due to other jobs in the system. This experiment used 2 cores of an *Intel Xeon Gold 6448Y* CPU and one *NVIDIA L40s* GPU. Correspondingly, the IQM when evaluating `AgarCL` without a GPU was 1,440 and 1,092, respectively.

## D  IMPLEMENTATION DETAILS IN AGARCL

In this section, we further discuss the details of the full-game experiment and the implementation of the fixed-policy bots.

### D.1  AGARCL EXPERIMENT DETAILS

In our main evaluation, we use a $128 \times 128$ arena containing 500 pellets, 10 viruses, and 8 heuristic bots. Upon being consumed, any agent or bot respawns immediately with an initial mass of 25, while all other entities preserve their current state. We describe the bot heuristics in Appendix D.2.

Every 600 ticks (150 ticks if we use frame skipping of 4), pellets and viruses that have been consumed or destroyed are regenerated with a uniform probability across the arena. This maintains fixed totals of 500 and 10, respectively. Mass decay is applied at intervals of 60 ticks. We conduct each trial for $160 \times 10^6$ frames.

### D.2  FIXED-POLICY BOTS

We implemented four heuristic bots in `AgarCL`: Aggressive, Aggressive-Shy, Hungry, and Hungry-Shy. We did so to explore how simple rules shape emergent game dynamics. All bots

follow a fixed policy that always targets the nearest pellet but differ in their interactions with other agents/bots. This set of behaviours allows us to assess how different action priors influence both individual success and the overall balance of the game (see Appendix I.3). The four bots' policies are:

- Aggressive: This bot first looks for any smaller opponent within a defined radius and attempts to consume it; if no suitable target is found, it switches to pellet collection.
- Aggressive-Shy: Like the Aggressive bot, it will hunt smaller opponents, but if a larger opponent approaches within its "shy" radius, it immediately flees and only returns to hunting once the threat has passed.
- Hungry: This bot ignores other players entirely and chases the closest pellet at every step.
- Hungry-Shy: Focused on pellet foraging like the Hungry bot, it additionally monitors for larger opponents: if one comes too close, it retreats before resuming its hunt.

## E  COMPUTATIONAL RESOURCES

To thoroughly evaluate our approach, we conducted an extensive hyperparameter tuning through grid search, running over a thousand individual experiments to identify high-performing configurations for each task. While this ensured strong performance, it came at a significant computational cost, a limitation worth noting for future work and reproducibility.

Table 1: Compute resource utilization across four clusters.

| Cluster | Elapsed (days) | Total CPU Usage (years) | Total GPU Usage (years) | Memory (32 GB units, years) |
|---|---|---|---|---|
| Cluster 1 | 3435.0 | 20.12 | 9.40 | 301.2 |
| Cluster 2 | 1162.0 | 10.20 | 3.16 | 96.0 |
| Cluster 3 | 162.9 | 1.65 | 0.44 | 14.3 |
| Cluster 4 | 884.0 | 7.76 | 2.42 | 77.6 |
| **Total** | **5644.0** | **39.73** | **15.42** | **489.1** |

**Cluster hardware details:**

- **Cluster 1:** Intel Xeon Gold 6448Y CPU; NVIDIA L40s (48GB) GPU.
- **Cluster 2:** Intel E5-2683 v4, Silver 4216, Platinum 8160F/8260 CPUs; NVIDIA P100 (12 GB), V100 (32 GB) GPUs.
- **Cluster 3:** Intel Xeon Gold 6238/6248, Silver 4110, AMD EPYC 7742/7713 CPUs; NVIDIA P100 (12 GB), V100 (16/32 GB), T4 (16 GB), A100, RTX A5000 GPUs.
- **Cluster 4:** Intel Xeon Gold 6148 CPU; NVIDIA V100SXM2 (16 GB) GPU.

## F  ADDITIONAL RELATED WORK

**Continual Reinforcement Learning.**  There have been many attempts to formalize the continual reinforcement learning (CRL) problem, each highlighting different aspects of non-stationarity and lifelong adaptation. Khetarpal et al. (2022) introduce a taxonomy for continual reinforcement learning (CRL) by focusing on two fundamental dimensions of non-stationarity: 'scope', referring to the extent of variation in tasks or domains, and 'driver', representing the main source of change. Building on this, Abel et al. (2023) define CRL as an ongoing adaptation process, in which agents continuously update their policies in response to evolving objectives, dynamics, or reward structures. More recently, Kumar et al. (2025) define continual learning under computation constraints by proposing a framework that preserves the balance of past knowledge with the efficiency of online updates over an extended period. Collectively, these CRL formalisms establish foundational principles that guide the design of benchmarks and algorithms.

**Existing Environments.**  Traditional RL testbeds (e.g., Zakka et al., 2025; Tassa et al., 2018) have driven rapid progress in RL, but they are not tailored for CRL. Accordingly, many CRL studies resort to switching among different games to induce non-stationarity (e.g., Abbas et al., 2023), relying on clearly defined train-test boundaries and assuming a well-structured notion of tasks and episodes.

However, such an approach is quite often contrived and, arguably, artificial. As we advocate in this paper, we believe slower and smoother changes are much more representative of the non-stationarity in the problems used to motivate CRL. JellyBeanWorld (Platanios et al., 2020) is the main platform for CRL in non-episodic settings that we are aware of. In JellyBeanWorld, agents navigate an infinite two-dimensional grid, interacting with various items by collecting or avoiding them. The agent's states are partially observable. Although JellyBeanWorld is clearly valuable for CRL research, it has simpler observations and (discrete) dynamics, without the ever-changing nature of AgarCL.

**Agar-Like Environments.** Most of the results relying on Agar-like environments for evaluation consisted of assessing basic agent capabilities, mostly related to pellet eating (Wiehe et al., 2018; Ansó et al., 2019), without ever putting forward the environment as a key artifact. To the best of our knowledge, GoBigger (Zhang et al., 2023) is the only other actual evaluation framework for AI research based on the Agar.io game. It shares certain surface-level similarities with our own, particularly in its Agar-style gameplay mechanics, but it was introduced to support a fundamentally different problem. Its main goal is to provide a platform to study collective behaviours in multi-agent reinforcement learning. Thus, its features consist of supporting multiple teams of agents (in the original game, collaboration should emerge from communication) in settings with agents organized into a few teams. The larger environment configuration supports 24 agents organized into four teams (6 each). The game is designed to be *episodic*, as the larger maps artificially have episodes that have at most $14,400$ frames (12 minutes). These are all in direct contrast to the continual learning problem we focus on, with potentially unbounded episodes and no pre-defined teams, but with a much bigger number of agents. GoBigger also does not support pixel-based observations; it just supports something akin to our symbolic observations with information about objects' positions and velocities. Finally, we benchmarked the frames per second (FPS) over ten independent trials using the `st_t6p4`[9] configuration—the largest map setting in the GoBigger implementation. Our environment significantly outperforms GoBigger in simulation speed, achieving an interquartile mean (IQM) of $4,212$ fps with GoBigger-style observations, compared to GoBigger's IQM of 205 fps under the same observation setup. This experiment was run using 2 cores of an *Intel Xeon Gold 6448Y* CPU.

# G ALGORITHM DETAILS

In this section, we focus on the adaptations we have made to DQN, PPO, and SAC to allow them to work somewhat effectively in `AgarCL`.

**DQN.** We used PFRL's (Fujita et al., 2021) DQN implementation. We did so by discretizing the continuous actions in the environment. We ended up with 24 actions: 8 directions times the 3 discrete actions. The predefined directions were: UP $(0, 1)$, UP-RIGHT $(1, 1)$, RIGHT $(1, 0)$, DOWN-RIGHT $(1, -1)$, DOWN $(0, -1)$, DOWN-LEFT $(-1, -1)$, LEFT $(-1, 0)$, and UP-LEFT $(-1, 1)$.

As discussed throughout, the resets inherent to episodic tasks can be beneficial for exploration by allowing agents to recover from "bad" states. However, in continual problems, the agent must naturally recover from a "bad" state. In our experiments, we noticed that the $\epsilon$-greedy strategy was ineffective in the non-episodic tasks we considered. Thus, in such tasks, we introduced temporally-extended exploration (Machado, 2019) through $\epsilon z$-greedy (Dabney et al., 2021), which achieves temporal persistence by extending the random actions in an exploratory step over multiple steps using a heavy-tailed duration distribution.

**PPO.** Our implementation extends PFRL's to the hybrid action setting. We adopted a shared neural network with two heads: an *actor* head that outputs action probabilities and a value *critic* head that estimates state values. In preliminary results, we observed that this architecture was more effective than having two independent networks. The actor head splits a 256-dimensional feature vector into discrete and continuous branches: a softmax head for categorical action probabilities and a Gaussian head (state-independent covariance) for continuous action means and variances, sampling both at each timestep to form a factored joint policy. The critic head is a compact one-layer MLP producing a scalar $V(s)$, trained with PPO's clipped value-function loss and generalized advantage estimation. Finally, we relied on PPO's entropy regularization for exploration.

---

[9]This configuration typically includes 1000 pellets, a $144 \times 144$ arena, and 14,400 frames per episode. We modified the GoBigger implementation to support a single player.

**SAC.** Similarly to PPO, we extended PFRL's SAC implementation to the hybrid action setting, and we relied on the algorithm's entropy maximization term for exploration. Unlike PPO's single value head, our implementation comprises three fully independent networks—one actor and two critics—each with its own encoder, $\phi$, that directly processes the raw observation. The actor mirrors the PPO policy architecture but with an entropy regularized objective. Likewise, each critic encodes the observation using $\phi$. The resulting representation is then concatenated with the continuous action, after which the critic predicts Q-values for each discrete action. The Q-values are selected with respect to the actor's sampled discrete action. By structuring the critics this way we avoid the combinatorial explosion that would arise if we naively input every possible hybrid action pair, and we prevent gradient interference between discrete and continuous parameters.

## H   Tuning Details

Hyperparameter tuning is especially critical in reinforcement learning due to the inherent instability and sensitivity of RL algorithms. Unlike supervised learning, RL involves exploration-exploitation trade-offs, nonstationary data distributions, and delayed rewards, all of which can magnify the effects of poorly chosen hyperparameters. Parameters such as step size, discount factor, exploration noise, entropy regularization (in policy gradient methods), and update frequencies can drastically influence the learning dynamics and final policy performance. Improper tuning can lead to divergence, suboptimal policies, or excessive variance in performance.

### H.1   DQN Tuning

For tuning DQN, we swept over the hyperparameters listed in Table 2. As discussed in the paper, both the agent's network and target network consist of three convolutional layers, each followed by a ReLU activation and a Layer Normalization layer. In preliminary experiments, we found that omitting Layer Normalization prevented the agent from learning altogether. All weights were initialized using a LeCun Normal Initialization LeCun et al. (2012). Each hyperparameter combination was trained using three different random seeds. After identifying the best hyperparameter configuration, we conducted ten additional independent runs using the best-performing configuration (see Table 3 on the next page). As shown in Table 2, also on the next page, the discount factor, $\gamma$, the number of epochs, the replay buffer size, and the target-network update interval were each fixed at a single value, while the remaining hyperparameters were swept. Finally, we applied the hyperparameter settings from continual MINI-GAME 4 to train the full-game agent.

Table 2: Values of hyperparameters that we swept over when tuning DQN.

| Hyperparameter | Values / Settings |
|---|---|
| Step size (-lr) | $10^{-5}, 3 \cdot 10^{-5}, 10^{-4}, 3 \cdot 10^{-4}$ |
| Batch Accumulator (-batch_accumulator) | "sum", "mean" |
| Soft Update Coefficient (-tau) | $10^{-2}, 5^{-3}$ |
| Batch Size (-batch-size) | 32, 64 |
| Replay Buffer Size (-replay-buffer) | $10^5$ |
| Number of Epochs (-epochs) | 1 |
| Target Network Update Interval (-update_interval) | 4 |
| Gamma (-$\gamma$) | 0.99 |
| Exploration algorithm | $\epsilon$-Greedy, $\epsilon z$-Greedy |

Table 3: Best hyperparameters for DQN on each minigame.

| Category | Mini-game | Hyperparameters | | | | Exploration Algorithm |
|---|---|---|---|---|---|---|
| | | Step Size | Batch Accumulator | Soft Update Coefficient | Batch Size | |
| Episodic | ① | $10^{-4}$ | mean | $5 \cdot 10^{-5}$ | 64 | $\epsilon$-Greedy |
| | ② | $3 \cdot 10^{-4}$ | sum | $5 \cdot 10^{-5}$ | 32 | $\epsilon$-Greedy |
| | ③ | $10^{-4}$ | mean | $5 \cdot 10^{-5}$ | 32 | $\epsilon$-Greedy |
| | ④ | $3 \cdot 10^{-4}$ | sum | $10^{-2}$ | 32 | $\epsilon$-Greedy |
| | ⑤ | $10^{-5}$ | mean | $10^{-2}$ | 64 | $\epsilon$-Greedy |
| | ⑥ | $10^{-5}$ | mean | $10^{-2}$ | 64 | $\epsilon$-Greedy |
| Continuing | ① | $10^{-4}$ | sum | $5 \cdot 10^{-5}$ | 32 | $\epsilon$z-Greedy |
| | ② | $3 \cdot 10^{-4}$ | sum | $10^{-2}$ | 32 | $\epsilon$z-Greedy |
| | ③ | $10^{-4}$ | mean | $5 \cdot 10^{-5}$ | 32 | $\epsilon$z-Greedy |
| | ④ | $10^{-4}$ | sum | $5 \cdot 10^{-5}$ | 32 | $\epsilon$z-Greedy |
| | ⑤ | $10^{-4}$ | mean | $10^{-2}$ | 32 | $\epsilon$z-Greedy |
| | ⑥ | $3 \cdot 10^{-4}$ | mean | $10^{-2}$ | 32 | $\epsilon$z-Greedy |
| Other Agents | ⑦-Small-Sparse[1] | $10^{-5}$ | mean | $5 \times 10^{-2}$ | 32 | $\epsilon$-Greedy |
| | ⑦-Normal-Dense[2] | $3 \times 10^{-4}$ | mean | $10^{-2}$ | 64 | $\epsilon$-Greedy |
| | ⑧-Small-Sparse[1] | $10^{-5}$ | mean | $5 \times 10^{-2}$ | 64 | $\epsilon$-Greedy |
| | ⑧-Normal-Dense[2] | $1 \times 10^{-5}$ | sum | $10^{-2}$ | 64 | $\epsilon$-Greedy |
| Virus | ⑨ | $3 \times 10^{-4}$ | sum | $10^{-2}$ | 128 | $\epsilon$-Greedy |

[1] Limited (Sparse) arena: $200 \times 200$ with 200 randomly respawned pellets.
[2] Normal (Dense) arena: $350 \times 350$ with 500 randomly respawned pellets.

## H.2 PPO TUNING

Hyperparameter tuning for PPO is particularly challenging due to the size of the configuration space: minimally covering relevant combinations requires at least 324 runs per experiment. We employed a shared neural network to learn both actor and critic, using the same architecture described in Appendix G. All network weights are initialized using LeCun normal initialization (LeCun et al., 2012). The policy head is a Gaussian distribution with a state-independent, learned covariance—a configuration that is standard in PPO implementations (e.g., Henderson et al., 2018).

In Table 4, we have the choice between two reward normalization schemes as a hyperparameter. The first is *min–max normalization*, which linearly rescales the raw reward $r_t$ into the range $[-1, 1]$:

$$\tilde{r}_t^{(\text{min–max})} = \frac{r_t - r_{\min}}{r_{\max} - r_{\min} + \epsilon}, \tag{2}$$

where $\epsilon > 0$ is a small constant added for numerical stability.

The second method, *variance normalization*, uses an exp. weighted moving average of the returns:

$$G_t = \gamma\, G_{t-1} + r_t, \tag{3} \qquad \tilde{r}_t^{(\text{var–norm})} = \frac{r_t}{\sqrt{\text{Var}[G_t] + \epsilon}}, \tag{4}$$

where $G_t$ is the smoothed return, and normalization is performed by dividing the reward by the square root of its running variance. This ensures that the normalized reward maintains approximately unit variance under an exponential moving average with discount factor $\gamma$. After normalization, the reward is **clipped** to lie within the fixed range $[-10, 10]$ to limit the effect of outliers during training:

$$\tilde{r}_t^{(\text{clipped})} = \text{clip}\left(\tilde{r}_t^{(\text{var–norm})}, -10, 10\right). \tag{5}$$

Initially, we swept over different hyperparameter combinations for PPO in some episodic settings following Table 4. Through these experiments, we observed that a value function coefficient of 0.9, an update interval of 5000, step sizes of either $10^{-5}$ or $3 \times 10^{-5}$, and epochs set to either 10 or 15 yielded consistently strong performance. Based on sensitivity analyses across most mini-game tasks, we narrowed down the range of hyperparameters we swept over for the other tasks. The smaller set of hyperparameters we swept over is shown in Table 5. Table 6 shows the best-performing hyperparameters that we used in each task at the end. Finally, we applied the hyperparameter settings from continual MINI-GAME 4 to train the full-game agent.

Table 4: Values of hyperparameters that we swept over when tuning PPO.

| Hyperparameter | Values / Settings |
|---|---|
| Reward function (`-reward`) | `min_max`, `variance_norm` |
| Step size (`-lr`) | $10^{-5}, 3 \times 10^{-5}, 3 \times 10^{-4}, 10^{-4}$ |
| Epochs (`-epochs`) | 10, 15, 20 |
| Max gradient norm (`-max-grad-norm`) | 0.5, 0.7, 0.9 |
| Entropy coefficient (`-entropy-coef`) | 0.05, 0.01, 0.1, 0.5 |
| Clipping epsilon (`-clip-eps`) | 0.2, 0.4 |
| Discount factor (`-gamma`) | 0.995 |
| GAE parameter (`-lambda`) | 0.97 |
| Value-function coefficient (`-value-func-coef`) | 0.9, 0.5 |
| Batch size (`-batch-size`) | 64 |
| Update interval (`-update-interval`) | 1024, 2048, 5000 |

Table 5: Updated PPO hyperparameters used for tuning.

| Hyperparameter | Values / Settings |
|---|---|
| Reward function (`-reward`) | `min_max`, `variance_norm` |
| Step size (`-lr`) | $10^{-5}, 3 \times 10^{-5}$ |
| Epochs (`-epochs`) | 10, 15 |
| Max gradient norm (`-max-grad-norm`) | 0.5, 0.7, 0.9 |
| Entropy coefficient (`-entropy-coef`) | 0.05, 0.01, 0.1, 0.5 |
| Clipping epsilon (`-clip-eps`) | 0.2, 0.4 |
| Discount factor (`-gamma`) | 0.995 |
| GAE parameter (`-lambda`) | 0.97 |
| Value-function coefficient (`-value-func-coef`) | 0.9 |
| Batch size (`-batch-size`) | 64 |
| Update interval (`-update-interval`) | 5000 |

Table 6: Best hyperparameters for PPO on each minigame.

| Category | Mini-game | Hyperparameters | | | | | |
|---|---|---|---|---|---|---|---|
| | | Reward Function | Step Size | Epochs | Max Grad Norm | Entropy Coef | Clip Eps |
| Episodic | ① | Min Max | $10^{-4}$ | 15 | 0.7 | 0.01 | 0.4 |
| | ② | Min Max | $10^{-5}$ | 10 | 0.7 | 0.01 | 0.2 |
| | ③ | Min Max | $10^{-5}$ | 10 | 0.9 | 0.01 | 0.2 |
| | ④ | Min Max | $10^{-5}$ | 15 | 0.9 | 0.05 | 0.4 |
| | ⑤ | Min Max | $10^{-4}$ | 10 | 0.7 | 0.01 | 0.4 |
| | ⑥ | Min Max | $3 \times 10^{-5}$ | 10 | 0.5 | 0.05 | 0.2 |
| Continuing | ① | Min Max | $10^{-4}$ | 4 | 0.5 | 0.005 | 0.2 |
| | ② | Variance Norm | $10^{-5}$ | 10 | 0.5 | 0.05 | 0.4 |
| | ③ | Min Max | $10^{-4}$ | 4 | 0.5 | 0.01 | 0.3 |
| | ④ | Variance Norm | $10^{-5}$ | 10 | 0.5 | 0.1 | 0.4 |
| | ⑤ | Min Max | $3 \times 10^{-5}$ | 10 | 0.7 | 0.05 | 0.4 |
| | ⑥ | Min Max | $3 \times 10^{-4}$ | 15 | 0.7 | 0.05 | 0.2 |
| Other Agents | ⑦-Small-Sparse[1] | Min Max | $3 \times 10^{-5}$ | 10 | 0.5 | 0.05 | 0.4 |
| | ⑦-Normal-Dense[2] | Variance Norm | $10^{-5}$ | 10 | 0.9 | 0.1 | 0.4 |
| | ⑧-Small-Sparse[1] | Min Max | $3 \times 10^{-5}$ | 10 | 0.5 | 0.05 | 0.2 |
| | ⑧-Normal-Dense[2] | Min-Max | $3 \times 10^{-5}$ | 10 | 0.7 | 0.05 | 0.2 |
| Virus | ⑨ | Min Max | $10^{-5}$ | 10 | 0.7 | 0.01 | 0.4 |

[1] Uses a $200 \times 200$ arena with 200 randomly respawned pellets.
[2] Uses a $350 \times 350$ arena with 500 randomly respawned pellets.

## H.3 SAC TUNING

We swept over the hyperparameter combinations in Table 7. In SAC, we employed three separate networks. All network weights were initialized using LeCun normal initialization(LeCun et al., 2012). Table 8 summarizes the best-performing hyperparameters across our tasks. Also, we applied the hyperparameter settings from continual MINI-GAME 4 to train the full-game agent.

Table 7: Values of hyperparameters that we swept over when tuning SAC.

| Hyperparameter | Values / Settings |
|---|---|
| Step size (-lr) | $10^{-4}, 3 \times 10^{-5}, 10^{-5}$ |
| Reward function (-reward) | `min_max`, `variance_norm` |
| Replay buffer size (-replay-buffer) | $10^5$ |
| Soft update coefficient (-tau) | $10^{-2}, 5 \times 10^{-3}, 10^{-3}$ |
| Max gradient norm (-max-grad-norm) | 0.5, 0.7, 0.9 |
| Temperature step size (-temperature-lr) | $10^{-4}, 3 \times 10^{-4}$ |
| Update interval (-update-interval) | 4 |
| Batch size (-batch-size) | 64 |
| Discount factor (-gamma) | 0.99 |

Table 8: Best hyperparameters for SAC on each minigame.

| Category | Mini-game | Hyperparameters | | | | |
|---|---|---|---|---|---|---|
| | | Reward Function | Step Size | Soft Update Coefficient | Temperature LR | Max Grad Norm |
| Episodic | ① | Min Max | $3 \times 10^{-5}$ | 0.001 | $10^{-4}$ | 0.7 |
| | ② | Min Max | $3 \times 10^{-5}$ | 0.001 | $10^{-4}$ | 0.7 |
| | ③ | Min Max | $10^{-4}$ | 0.01 | $10^{-4}$ | 0.7 |
| | ④ | Min Max | $10^{-5}$ | 0.001 | $10^{-4}$ | 0.7 |
| | ⑤ | Min Max | $10^{-5}$ | 0.001 | $10^{-4}$ | 0.7 |
| | ⑥ | Min Max | $10^{-4}$ | 0.001 | $10^{-4}$ | 0.5 |
| Continuing | ① | Variance Norm | $3 \times 10^{-5}$ | 0.001 | $10^{-4}$ | 0.5 |
| | ② | Variance Norm | $3 \times 10^{-5}$ | 0.001 | $10^{-4}$ | 0.7 |
| | ③ | Variance Norm | $3 \times 10^{-5}$ | 0.005 | $10^{-4}$ | 0.9 |
| | ④ | Variance Norm | $10^{-5}$ | 0.01 | $10^{-4}$ | 0.5 |
| | ⑤ | Variance Norm | $10^{-5}$ | 0.001 | $10^{-4}$ | 0.7 |
| | ⑥ | Variance Norm | $3 \times 10^{-5}$ | 0.005 | $10^{-5}$ | 0.9 |
| Other Agents | ⑦-Small-Sparse[1] | Variance Norm | $10^{-4}$ | 0.01 | $10^{-4}$ | 0.9 |
| | ⑦-Normal-Dense[2] | Variance Norm | $10^{-4}$ | 0.01 | $10^{-4}$ | 0.7 |
| | ⑧-Small-Sparse[1] | Variance Norm | $10^{-5}$ | 0.001 | $10^{-4}$ | 0.9 |
| | ⑧-Normal-Dense[2] | Variance Norm | $10^{-5}$ | 0.005 | $10^{-4}$ | 0.9 |
| Virus | ⑨ | Min Max | $10^{-5}$ | 0.01 | $10^{-4}$ | 0.7 |

[1] Sparse setting: $200 \times 200$ arena with 200 pellets.
[2] Dense setting: $350 \times 350$ arena with 500 pellets.

## I MINI-GAME RESULTS

In this section, we provide visualizations of the mini-games used in our evaluation (Section I.1) and the complete set of results mentioned in the main paper. Those include results in the continual pellet collection tasks (Section I.2), including analyses over the algorithms hyperparameter sensitivity (Section I.2.1) and results in those mini-games when augmenting PPO with a GRU (Section I.2.2). Additionally, we report results in the mini-games in which the agent was faced with another bot (Section I.3) and those in which it was expected to successfully interact with viruses (Section I.4).

### I.1 ILLUSTRATIVE FIGURES OF THE PELLET-COLLECTION MINI-GAMES

The mini-games for pellet collection are divided into two sets: Square and Random. The Square mini-games require the agent to collect pellets along a square-shaped path, and it has three versions. The simplest version involves collecting pellets only, with no additional challenges. The second

introduces mass decay, and in the third, the agent starts with a much bigger mass (1000 instead of 25). The set of mini-games with randomly scattered pellets uses the same variations. New pellets appear every 600 environment ticks. Figure 8 depicts all these variants.

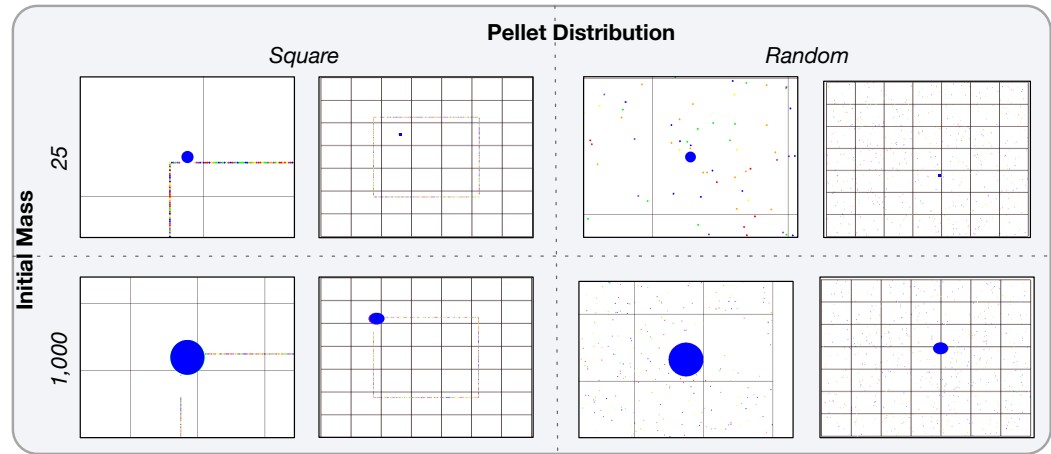

Figure 8: Pellet Collection mini games. They were defined in terms of the pellet distribution in the arena, the agent's initial mass, and whether the agent's mass decays (which is hard to depict in an image). Note that we do not have eight mini-games because starting with a mass of 1,000 and no mass decay is an uninteresting setting. In each quadrant, the agent's actual view at the beginning of the mini-game is shown on the left, and, to provide a sense of scale, a zoomed-out perspective of the same setting is shown on the right. The agent is depicted in blue.

## I.2 CONTINUAL PROBLEMS

We discussed this setting in detail in the main paper; however, due to space constraints, we were unable to present the complete set of results there. They are available in Figure 9 below.

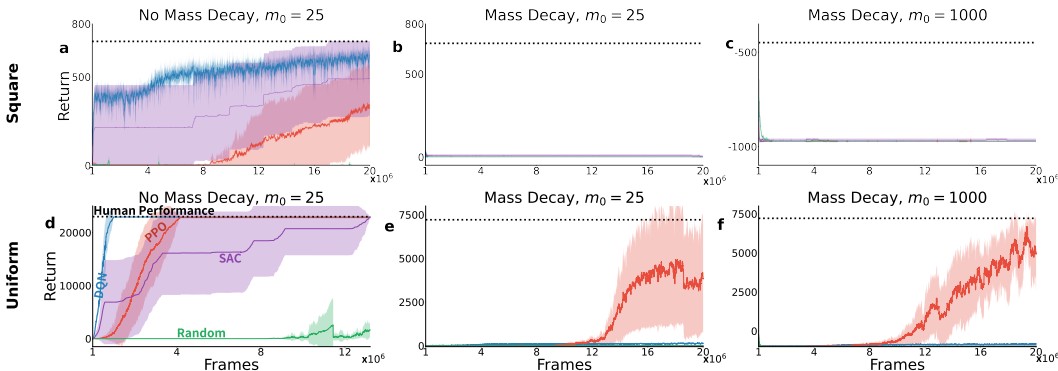

Figure 9: Performance of RL methods on *continual* pellet-collection mini-games. Panels ⓐ, ⓑ, and ⓒ show the performance on *the square-path tasks* (mini-games 1, 2, and 3), while ⓓ, ⓔ, and ⓕ show the performance on *randomly regenerated tasks* (mini-games 4, 5, and 6). Note that the y-axis scales vary across plots. The dashed line marks human performance, and the green line marks the random policy. The shaded region shows the 95% CI over 10 runs, computed using the t-distribution.

### I.2.1 HYPERPARAMETER SENSITIVITY ACROSS MINI-GAMES

We ran thousands of GPU jobs to tune and evaluate, on a per-environment basis, the baselines discussed in the previous section. Hyperparameter tuning is a major challenge in RL, and an even bigger one in continual RL. Each algorithm is impacted differently by each hyperparameter. To test

robustness, we used hyperparameters tuned in one mini-game to evaluate performance in another. In many tasks, PPO tuned on a particular mini-game often collapses when applied to a different mini-game. For SAC, using hyperparameters tuned for different mini-games sometimes yields better results than the hyperparameters optimized for the task itself! DQN, on the other hand, demonstrates surprising robustness—cross-task hyperparameter transfers have minimal impact on its peak performance. This big variability can be due to the complexities in hyperparameter selection strategies (Patterson et al., 2024), including the fact that we were able to afford only three seeds per configuration, which is certainly not enough for an accurate estimate. This is another common practice in the field that can be quite detrimental. The actual plots for these results are in Figure 10. These results underscore a critical insight: no single hyperparameter setting is robust across all tasks.

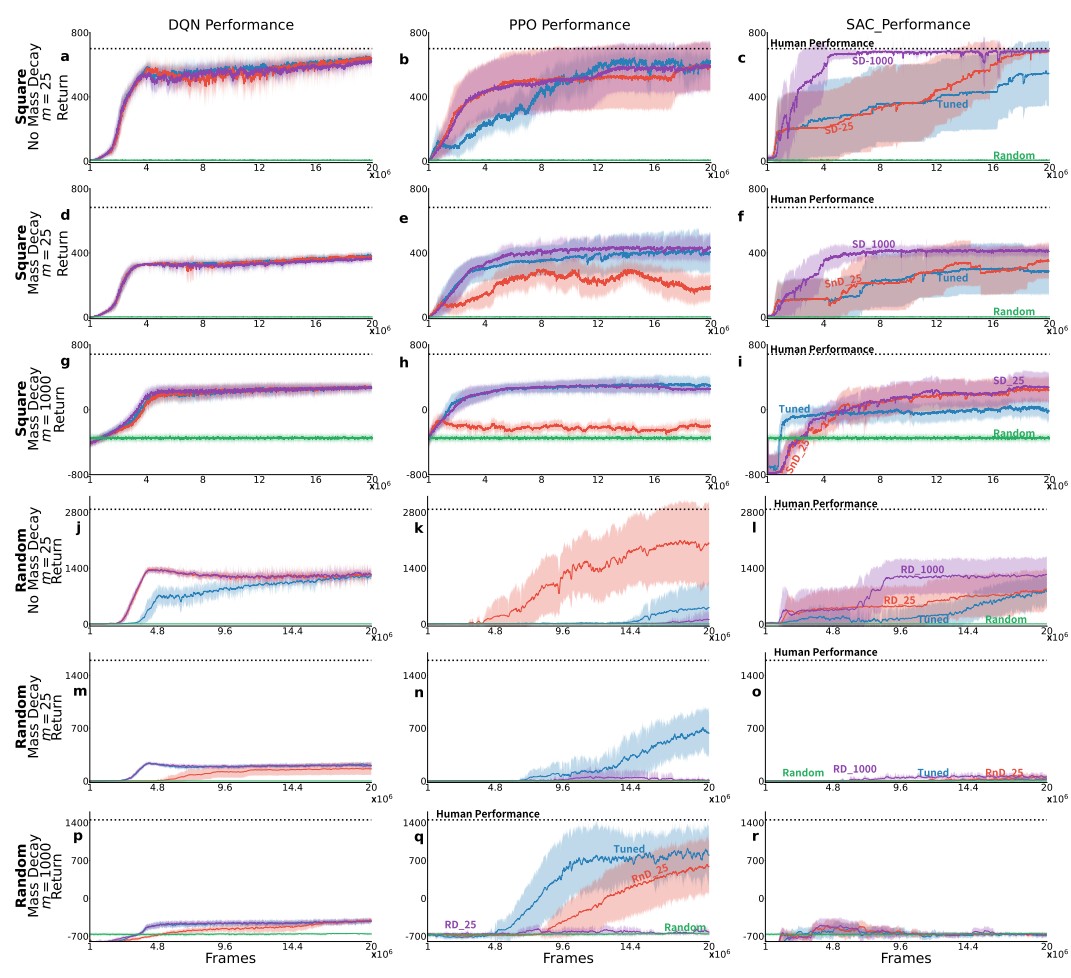

Figure 10: Performance of DQN, PPO, and SAC on six mini-games using cross-evaluated hyperparameters. The mini-games are grouped into two categories: the top three rows correspond to the **square path** group (Square-path pellet collection tasks), and the bottom three rows to the **random** group (randomly regenerated pellet tasks). For each mini-game, agents are evaluated not only using their own tuned hyperparameters but also using the hyperparameters optimized for the other two tasks within the same group. Each row shows the performance of a baseline algorithm across three hyperparameter configurations. For example, panels ⓐ, ⓑ, and ⓒ show DQN, PPO, and SAC in MINI-GAME 1 under different hyperparameter settings. The naming convention is as follows: `SD-1000` denotes a `Square` path setting with mass `Decay` and an initial mass of 1000, while `SnD-25` indicates `no Decay` and an initial mass of 25. `Tuned` is considered for the best hyperparameter on a particular mini-game. We use $R$ instead of $S$ on the panels for the mini-games in which pellets were randomly spread in the environment (instead of a square).

### I.2.2 PPO AUGMENTED WITH A GRU

One might wonder whether approaches able to tackle partial observability would be effective in `AgarCL`, mainly in the settings with a high degree of partial observability. As previously discussed, PPO is the best-performing algorithm among those we considered; because of that, we decided to evaluate PPO, augmented with a recurrent network, in the pellet collection MINI-GAMES.

Specifically, we evaluated PPO with Gated Recurrent Units (GRUs; Cho et al., 2014). The model comprises three convolutional layers, followed by a 256-unit GRU, and a final linear layer that splits into actor and critic heads. We re-tuned all hyperparameters for this new algorithm. All other experimental procedures were kept the same.

As shown in Figure 11, GRUs did not lead to much improvement. It performed well in relatively simple scenarios as MINI-GAME 1 (see panel ⓐ), where it achieved human-level performance. However, it struggled in more challenging tasks, such as mini-games 2 and 3 (panels ⓑ and ⓒ). Notably, in the uniform variation (panels ⓓ-ⓕ), the performance of PPO with GRUs was even worse than that of the non-recurrent version. Table 9 reports the equivalent numerical results.

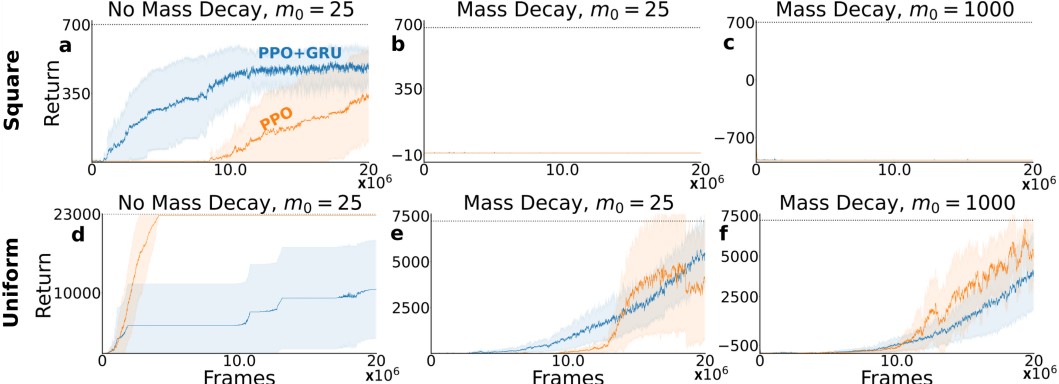

Figure 11: Performance of PPO+GRU and PPO on *continual* pellet-collection mini-games. Panels ⓐ, ⓑ, and ⓒ correspond to the *square-path* tasks (with no mass decay, mass decay $m_0 = 25$, and mass decay $m_0 = 1000$, respectively), while panels ⓓ, ⓔ, and ⓕ correspond to the *uniformly regenerated* pellets with the same decay settings. The shaded regions show the 95% CI over 10 evaluation runs, computed using the t-distribution. Note that the y-axis scales differ between panels.

Table 9: Performance of PPO with GRU-based recurrent architecture across six mini-games. Results are averaged over the last 100 data points. Values in parentheses denote standard deviations.

| Algorithm | Mini-Game 1 | Mini-Game 2 | Mini-Game 3 | Mini-Game 4 | Mini-Game 5 | Mini-Game 6 |
|---|---|---|---|---|---|---|
| PPO | 335 (3.0) | 0 (0) | −975 (0) | 22,463 (48.9) | 4,018 (126.8) | 5,064 (113.6) |
| PPO + RNN | 484 (10.0) | 0 (0) | −974 (0.3) | 20,533 (250) | 5,329 (30.9) | 4,036 (22.4) |

### I.3 INTERACTING WITH OTHER AGENTS

Aside from pellet collection, capturing (or fleeing) other agents is a key skill an agent is expected to have in `AgarCL`. In this section, we evaluate the agent in mini-games in which the agent is expected to interact with a single other bot in the environment. To make the problem easier for the agent, given the difficulty they faced in the continual mini-games, we conducted experiments in an episodic setting, where each episode terminates either when the agent is eaten or after 10,000 time steps, whichever occurs first. We conducted these experiments in two configurations, LARGE-DENSE setup and SMALL-SPARSE setup. We discuss both below.

### I.3.1 LARGE-DENSE SETUP

This first setup uses a standard number of pellets (500) and a normal-sized arena ($350 \times 350$). We used two types of fixed-policy bots: the *hungry* bot, which focuses solely on collecting pellets, and

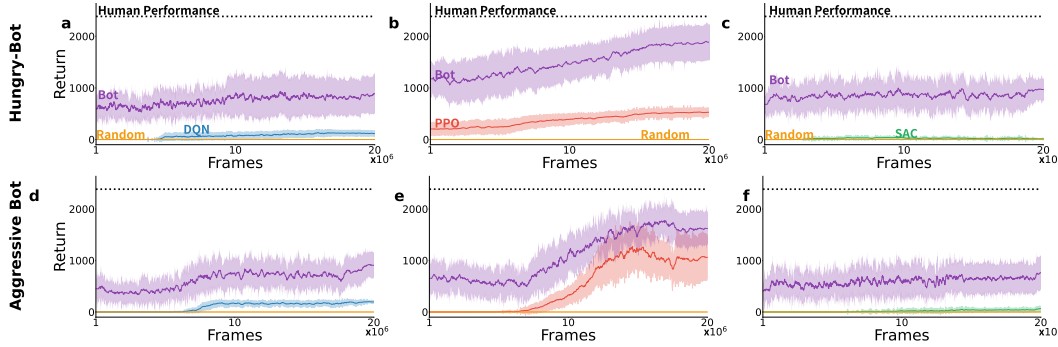

Figure 12: Performance of DQN, PPO, and SAC on *episodic* other-agent mini-games. Panels ⓐ and ⓓ show the performance of *DQN*, Panels ⓑ and ⓔ show the performance of *PPO*, and Panels ⓒ and ⓕ show the performance of *SAC*. The top row (ⓐ–ⓒ) corresponds to the *hungry-bot* task (mini-game 7), while the bottom row (ⓓ–ⓕ) corresponds to the *aggressive-bot* task (mini-game 8). Each experiment features a fixed-policy bot—either hungry or aggressive—in a standard-sized arena ($350 \times 350$) with 500 pellets. Both the agent and the fixed-policy bot start with an initial mass of 25. Shaded regions represent the 95% CI over 10 independent runs, computed using the $t$-distribution.

the *aggressive* bot, which prioritizes consuming any entity smaller than itself.

Figure 12 depicts the obtained results as well as the return obtained by the fixed-policy bot, as its return in comparison to the learning agent is quite informative. We can see that both DQN and SAC struggled in the presence of the aggressive and hungry bots. Gameplay videos recorded at various evaluation checkpoints revealed that these agents frequently became stuck in corners, a failure mode also observed in pellet collection mini-games. Consequently, the *aggressive* bot could consume the learning agents quickly, as reflected by the early plateau in their return curves in panels ⓓ and ⓕ. The performance of the *hungry* bot in panels ⓐ and ⓒ appears slightly better, focusing on collecting pellets rather than following the learning agents. PPO agent exhibited more robust behaviour. Panel ⓔ shows that PPO learned to avoid the aggressive bot effectively while collecting many pellets. This evasive strategy contributed to the steady increase observed in the agent's return, albeit at a slower rate than its adversary. However, in the hungry setting (panel ⓑ), PPO failed to consume the bot, even though the hungry bot did not attempt to attack it. This suggests that while PPO is effective in terms of survival, it may not exploit opportunities to eliminate non-aggressive opponents.

These results raise the question of whether PPO's success was due to the large arena and high pellet density. The SMALL-SPARSE setting, discussed next, addresses this question directly.

### I.3.2 SMALL-SPARSE SETUP

To limit the agent's ability to avoid interaction with bots, we evaluated them in a smaller arena ($200 \times 200$) with a reduced number of pellets (250). This configuration increases the likelihood of direct encounters between the learning agent and the fixed-policy bot.

Consistent with previous findings, the DQN and SAC baselines failed to demonstrate effective learning, irrespective of the type of bot in the environment (see Figure 13 on the next page). This is primarily attributable to recurring behavioural failures, such as becoming trapped in corners or failing to evade the bot, which are issues that persisted under these more restrictive conditions. Interestingly, the PPO agent also failed to learn in this environment, regardless of whether it was paired with the aggressive or the hungry bot. PPO could not consistently evade the aggressive bot or exploit the passive behaviour of the hungry one. In panel ⓔ, PPO's return clearly plateaus early, while the aggressive bot's return steadily increases, indicating its ability to repeatedly eat the agent with ease. Even more interesting is PPO's performance against the hungry bot. Despite the absence of an active threat, PPO failed to take advantage of the opportunity to survive and collect pellets. The consistent flattening of the agent's learning curves across all settings further reinforces this observation, indicating minimal policy improvement over time.

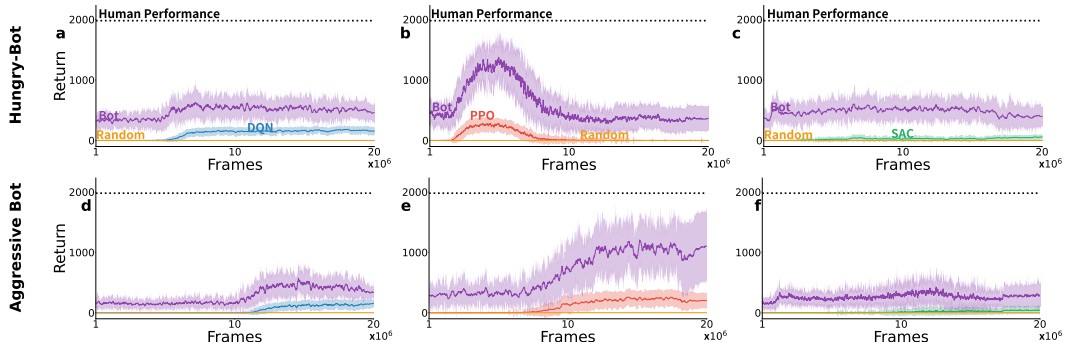

Figure 13: Performance of RL baselines—DQN, PPO, and SAC—on *episodic* other-agent mini-games. Panels ⓐ and ⓓ show the performance of *DQN*, Panels ⓑ and ⓔ show the performance of *PPO*, and Panels ⓒ and ⓕ show the performance of *SAC*. The top row (ⓐ–ⓒ) corresponds to the *hungry-bot* mini-game, while the bottom row (ⓓ–ⓕ) corresponds to the *aggressive-bot* mini-game. Each experiment features a fixed-policy bot—aggressive or hungry—in a limited arena ($200 \times 200$) with 250 pellets. The agent and the fixed-policy bot start with an initial mass of 25. Shaded regions represent the 95% confidence intervals over 10 independent runs, computed using the $t$-distribution.

## I.4 Interacting with Viruses

A final skill we probe for is whether agents can learn how to leverage viruses in the environment, in the most benign setting possible. The mini-game is configured to be fully observable. As illustrated in Figure 14, the opposing bot remains stationary, and a clear, linear arrangement of viruses is positioned between the agent and the bot. Notably, there are no pellets in this mini-game. The agent begins with a mass of 3000 and must engage with a larger, stationary bot that has a mass of 5000. To simplify dynamics further, mass decay is disabled. Training in this mini-game follows an episodic setup: an episode ends when one agent is eaten or after 1000 time steps.

The agent should learn to eject pellets toward the viruses, triggering them to split the stationary bot. The agent can then pass through the virus field and consume the smaller bot fragments. As shown in Figure 15, all learning agents failed to discover this strategy. Instead, they converged to a passive behaviour, remaining stationary for the entire episode. This highlights the difficulty these agents face in learning to leverage environmental elements, such as viruses, for strategic interaction, even in a simplified and fully observable setting.

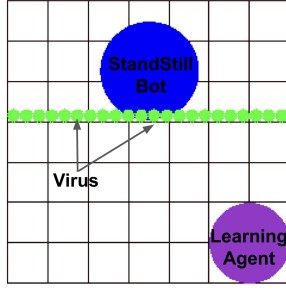

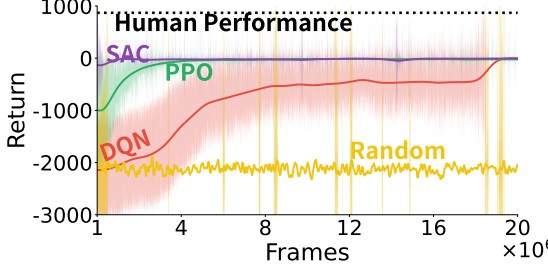

Figure 14: The learning agent (mass 3,000) is separated from a standstill bot (mass 5,000) by a line of static viruses (mass 100). The arena is fully observable with no pellets, mass decay, or virus respawning.

Figure 15: Performance of DQN, PPO, and SAC on the *virus-based* mini-game. Shaded regions show the 95% CI across 10 runs using the $t$-distribution.

## J    NUMERICAL RESULTS ON MINI-GAMES AND THE FULL GAME

Table 10 summarizes the results of the experiments conducted in this paper. Each reported value represents the final performance, calculated as the average reward over the last 100 evaluation episodes.

Table 10: Performance across `AgarCL` minigames averaged over 10 independent runs (std. dev. is reported between parentheses). Results are averaged over the last 100 data points: episodes in episodic tasks and 100 steps in continual tasks.

| Category | Mini-game | Scenarios | DQN | PPO | SAC | Human | Random |
|---|---|---|---|---|---|---|---|
| Pellet Collection (Episodic) | 1 | — | 642 (14.2) | 605 (25.3) | 546 (6.0) | 700 (0) | 4.81 (1.3) |
| | 2 | — | 382 (9.1) | 398 (12.9) | 285 (9.8) | 682 (0) | 0.81 (0.7) |
| | 3 | — | 271 (23.1) | 298 (28.8) | −16 (30.3) | 612 (0) | −356 (22.7) |
| | 4 | — | 1189 (44.2) | 391 (36.3) | 790 (98.6) | 2876 (0) | 4 (1.3) |
| | 5 | — | 214 (13.2) | 650 (58.9) | 16 (8.4) | 1600 (0) | 0.17 (0.16) |
| | 6 | — | −426 (20.0) | 812 (118.3) | −681 (12.8) | 1452 (0) | −657 (6.9) |
| Pellet Collection (Continual) | 1 | — | 619 (7.9) | 335 (3.0) | 419 (0.2) | 700 (0) | 0.01 (0.1) |
| | 2 | — | 0 (0) | 0 (0) | 2 (1.7) | 682 (0) | 0 (0) |
| | 3 | — | −975 (0) | −975 (0) | −975 (0) | −480 (0) | −975 (0) |
| | 4 | Sparse[1] | — | 22463 (48.9) | — | — | — |
| | | Dense[1] | 21402 (762.8) | 21970 (277) | 21997 (595.6) | 23000 (0) | 112.2 (12.5) |
| | 5 | Sparse | — | 144 (10.8) | — | — | — |
| | | Dense | 132 (9.3) | 4018 (126.8) | 0.012 (0.9) | 7215 (0) | 0.05 (0.09) |
| | 6 | Sparse | — | −618.3 (1.4) | — | — | — |
| | | Dense | −828 (10.5) | 5064.4 (113.6) | −968 (0.01) | 7215 (0) | −973 (1.1) |
| Other Agent (Episodic) | 7—Small, Sparse[2] | — | 157 (0.7) | 253 (35.6) | 31 (0.2) | 1980 (0) | 0.015 (0.04) |
| | 7—Large, Dense[2] | — | 146 (48.4) | 429.3 (136.4) | 25 (14.2) | 2385 (0) | 0.1 (0.16) |
| | 8—Small, Sparse | — | 189 (6.03) | 267 (26) | 114 (18.8) | 1800 (0.015) | 0.04 (0.04) |
| | 8—Large, Dense | — | 191 (14.56) | 1037 (55.2) | 69 (8.9) | 2035 (0.015) | 0.04 (0.04) |
| Virus (Episodic) | 9 | — | -36 (4.23) | -46 (6.45) | -5 (0.1) | 870 (20.5) | -2319 (8.94) |
| Full Game (Grand Arena) | — | — | 4 (3.1) | 8 (8.3) | 22 (11.1) | — | 0.06 (0.18) |

[1] In continuing pellet-collection tasks, "Sparse" refers to an arena with 250 pellets. "Dense" uses 500 pellets.
[2] "Small" arena: $200 \times 200$ with 200 pellets. "Large" arena: $350 \times 350$ with 500 pellets.

## K    HOW THE NUMBER OF PELLETS IMPACTS THE AGENT'S PERFORMANCE

In this section, we further evaluate how PPO behaves in more challenging pellet-collection mini-games, where the number of available pellets is significantly reduced.

In Figure 9, PPO demonstrates strong performance in collecting pellets, even in the continual setting. However, the task remains relatively easy due to the high number of pellets (500), which allows the agent to find and consume pellets with minimal effort, even in the presence of mass decay. The situation changes when the number of pellets is reduced to 250. Interestingly, PPO failed to learn effectively in MINI-GAMES ⑤ and ⑥, suggesting increased difficulty with exploration.

More generally, following the discussion on Section 4, in the context of designing easier settings in the full game such that we can see positive learning, we designed a variant of the environment in which the number of bots was de-

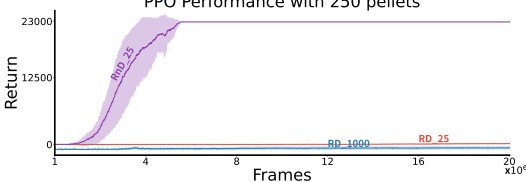

Figure 16:  PPO performance on *continuing randomly regenerated pellet-collection* in MINI-GAMES 4, 5, and 6.  `RD-1000` denotes a Randomly regenerated pellet setting with mass Decay and an initial mass of 1000; `SnD-25` indicates no Decay and an initial mass of 25. The plateau in the `RnD-25` curve after approx. 1M steps is due to the agent reaching the maximum cumulative reward in MINI-GAME ④.  Shaded regions indicate the 95% CI over 10 independent runs, using reference values from the t-table.

creased from 8 to 4, and we evaluated the impact of having fewer (400) or more (600, 1024) pellets than in the default setup. We maintained 10 viruses in the environment. We evaluated each agent for 86 million training steps.

As illustrated in Figure 17, increasing the number of pellets and reducing the number of bots does indeed make the task easier. Some of the curves depict a drop in performance later in training. While this is not surprising in continual RL due to issues such as loss of plasticity (Abbas et al., 2023; Lewandowski et al., 2023; Dohare et al., 2024), we did not investigate this phenomenon further, nor did we evaluate any mitigation strategies. The important observation, despite the high variance across runs, is that this setting makes the problem more tractable, so we can then consider the impact of freezing the agent's policy.

These results were the motivation for us to use $1,024$ pellets in the experiment in which we froze the agent's policy (in an easier environment, so we could first see learning).

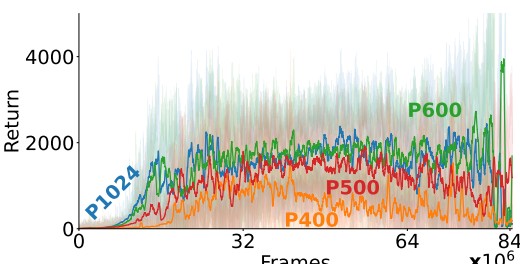

Figure 17: PPO performance across varying pellet densities within a simplified version of the default setup. All experiments use 10 viruses and 4 bots; with a varying number of pellets, which is labeled in each curve. For example, `P1024` denotes 1024 pellets. Results are averaged over 10 random seeds, and each curve is smoothed using a moving average with a window of 1000 steps.

## L CONTINUAL LEARNING IN AgarCL

In this section, we examine `AgarCL` from the perspective of continual RL. We first test the impact of freezing the agent's policy in a different setting; then, we evaluate the performance of PPO with continual backpropagation (Dohare et al., 2024), an approach explicitly designed for continual learning.

### L.1 FIXED POLICIES VS LEARNING AGENT IN AgarCL

In our earlier analysis, we identified performance degradation when training was interrupted at 32 million and 48 million steps in the setting with 8 bots and $1,024$ pellets. We also evaluated the impact of freezing the agent's policy in other scenarios. Specifically, we performed an additional experiment under the same conditions but with 4 bots and 500 pellets. We can also observe policy collapse for the agents frozen after 32M steps. We conjecture we would also observe policy collapse for the agent frozen after 48M steps if we waited long enough, but if not, this would be evidence that simpler settings (e.g., four bots) might make it easier for the agent to learn some non-trivial policy and maintain it, even though it is quite suboptimal.

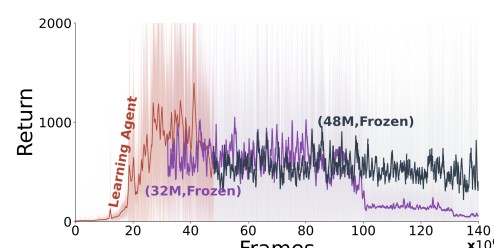

Figure 18: Performance of fixed-policy agents initialized from checkpoints at 32M and 48M steps. We report the moving average over 10 random seeds with a window size of 1000 steps.

### L.2 DETAILS ABOUT EXPERIMENTS WITH CONTINUAL LEARNING ALGORITHMS

The algorithms evaluated in Section 4.3 augmented PPO with Shrink and Perturb (Ash & Adams, 2020), ReDo (Sokar et al., 2023), or Continual Backpropagation (Dohare et al., 2024). In these three approaches, we used the same hyperparameters we used in PPO. We set the algorithm-specific hyperparameters according to the recommendations in the original papers. We detail those below. Co

**Shrink and Perturb.** We set $\alpha = 0.4$ (shrink) and $\beta = 0.2$ (perturb), so that, when appropriate, each parameter is updated according to $x' = 0.4\,x + 0.2\,x_0$, where $x$ is the current weight and $x_0$ is

a freshly initialized parameter. Shrink-and-Perturb is applied every 500,000 time steps.

**ReDo.** ReDo identifies and re-initializes inactive neurons in each linear or convolutional layer (excluding the output). For a representative minibatch, we record post-activation values, $a$, and compute per-unit activity scores, $s_j = \text{mean}(|a_j|)$, normalized as $\tilde{s}_j = s_j/(\bar{s} + 10^{-9})$. Units with $\tilde{s}_j \leq \tau$ are considered dormant. By default, we set $\tau = 0.05$.

ReDo (1) partially reinitializes incoming weights using LeCun normal or Kaiming uniform, (2) zeros outgoing connections to prevent contribution of dormant units, and (3) resets optimizer moments for affected parameters to avoid immediate re-amplification. The operation occurs every 2 million time steps and is computationally cheap compared with full retraining.

**Continual BackPropagation.** Continual Backpropagation (CB) tracks per-feature utility by monitoring activations and output weights, and periodically reinitializes features whose utility falls below a threshold. Replacement is performed via backward-pass hooks: fractional replacement counts accumulate until an integer replacement occurs, and reinitialized input weights are drawn uniformly within layer-specific bounds; corresponding output weights and normalization statistics are reset. The table 11 shows our default values.

Table 11: Continual Backpropagation hyperparameters.

| Layer type | Replacement rate | Maturity threshold |
|---|---|---|
| Convolutional | $1 \times 10^{-5}$ | 1000 steps |
| Linear | $1 \times 10^{-4}$ | 100 steps |

### L.3 FULL-GAME RESULTS WITH CONTINUAL LEARNING METHODS

Table 12: Full-Game performance over the last 100 steps, averaged across 10 runs. Mean and standard deviation are reported for each algorithm.

| Metric | Continual Backprop | ReDo | Shrink & Perturb | PPO |
|---|---|---|---|---|
| Mean | 3675.8 | 4233.9 | 2074.2 | 1706.53 |
| Std | 4381.81 | 5040.74 | 1868.18 | 2925.19 |

