# OpenReview forum: "The Cell Must Go On: Agar.io for Continual Reinforcement Learning"
_ICLR.cc/2026/Conference — Submitted to ICLR 2026_

### Official Review · Reviewer_YYvQ · 2025-10-27

**Soundness:** 3
**Presentation:** 3
**Contribution:** 2
**Rating:** 6
**Confidence:** 2

**Summary:**

The authors presented a new environment that is suitable for continual RL. It is based on a game and it supports a quite complex amount of states and actions. They benchmarked their environment over standard RL algorithms showing they are struggling to learn fixed policies even over simplified versions of the game.

**Strengths:**

- The paper is clear, and the description of the environment is detailed enough
- The advantage of the proposed framework over classical RL benchmarks is clear
- The information present in the main paper and in the appendix are complete and exaustive

**Weaknesses:**

- The relationships between the environment and the continual learning tasks are not stated explicitly until the experimental section
- The audience interested in this environment is really narrow
- The fact that most of the existing CL methods cannot be applied to this setting suggests that the research is currently on a different path than the one proposed by the authors

**Questions:**

1) You mentioned the time for training with SAC should be contextualized with information on the architecture you used in the main paper. You should also add some information about this also in the main paper.
2) Most of the experimental results are deferred to the appendix. I know there are space constraints, but at least some of the results from the continual learning setting should be included in the main paper.

---

> ### Author Response · Authors · 2025-11-18
>
> We thank the reviewer for noting that our paper introduces a new environment for continual learning research, providing clear benefits over existing literature in a comprehensive and exhaustive manner. We appreciate the opportunity to clarify the relationship to continual learning and address the concerns raised. That said, we were somewhat confused by the weaknesses mentioned and would like to better understand them. Given the positive overall assessment and the nature of the issues raised, we believe these points can be readily discussed to support a higher rating for our paper.
>
> > "The relationships between the environment and the continual learning tasks are not stated explicitly until the experimental section"
>
> We introduced AgarCL throughout the paper through a continual learning lens, emphasizing features that make it a good platform for continual learning research—such as its non-episodic interactions, partial observability, and smooth endogenous non-stationarity. Could the reviewer clarify which details they felt were only discussed in the experimental section? We are happy to bring these points forward, as we agree they should be stated early in the paper. One option is to provide some foreshadowing of our empirical results in the introduction; would this address the reviewer’s concern?
>
> > The audience interested in this environment is really narrow"
>
> We obviously cannot predict the impact our environment will have on the community, so we prefer to avoid a subjective discussion of the audience’s interest. That said, continual learning is a growing research area, with entire conferences, such as CoLLAs, dedicated to it. The features that define AgarCL, such as non-episodic, partially observable problems that continuously evolve, capture key properties of problems of interest. In many real-world systems, dynamics evolve continuously—for example, through wear-and-tear, drift in sensor statistics, or changes in operating conditions—rather than via discrete, externally defined task boundaries. Could the reviewer clarify what specific weakness is being raised here?
>
>
> > "You mentioned the time for training with SAC should be contextualized with information on the architecture you used in the main paper. You should also add some information about this also in the main paper."
>
> It is not clear to us what the reviewer is asking here. In footnote 6, we provide the actual runtimes for some of the runs, and architectural details are already included in Appendix G. Could the reviewer clarify their main concern? Is there specific information in the Appendix that you are suggesting we move into the main paper?
>
> > "Most of the experimental results are deferred to the appendix. I know there are space constraints, but at least some of the results from the continual learning setting should be included in the main paper."
>
> Yes, the page limit was the main reason we could not include additional results in the main paper. However, the key results are already discussed: Section 4.1 presents the performance of all algorithms on the main benchmark, Section 4.2 provides evidence supporting the claim that a fixed policy is less effective than a continually adapting one, and Section 4.3 highlights results from a specific mini-game to emphasize key aspects of the environment (with a summary of additional results in the appendix). The revised version we are allowed to submit during the rebuttal period includes one extra page, which will enable us to move more results from the appendix into the main paper. Specifically, we plan to move Figure 8, an expanded version of Section L.2, and part of Section I.2.2. We would be happy to hear if the reviewer has other suggestions.

---

> > ### Comment · Reviewer_YYvQ · 2025-11-28
> >
> > Dear authors,
> >  Thanks for the clarification and the further questions.
> > - regarding the "continual learning lens". I think it is a matter of readability. Perhaps the concepts you mentioned are present, but they may not always be straightforward to me.
> > - scope: This is not a weakness; it is a comment about the specific nature of the paper that will be of interest to the continual learning community.
> > - runtimes: this would be better. Again, providing running time without contextualizing the architecture is not meaningful to me.
> > - experiments: this was exactly my point. Having more results in the main paper can provide a clearer understanding of the capabilities of your framework.

---

### Official Review · Reviewer_Vnh3 · 2025-10-27

**Soundness:** 2
**Presentation:** 4
**Contribution:** 2
**Rating:** 2
**Confidence:** 4

**Summary:**

The paper introduces AgarCL, inspired by the Agar.io online game, positioned as an environment for studying continual RL. It features pixel-based observations, hybrid actions, and a non-episodic, long-horizon, partially observable setting. The objective is to grow by collecting pellets and consuming smaller opponents, while avoiding viruses and larger opponents. The authors evaluate standard RL algorithms (DQN, PPO, SAC) and find that none achieve strong performance, highlighting the environment’s difficulty. They also propose a suite of mini-games to isolate sub-challenges such as exploration and mass dynamics.

**Strengths:**

1. The rewards and hybrid action space (mimicking cursor control with discrete actions) are well designed and align with the gameplay dynamics.
2. The accompanying video provides a clear, intuitive overview of the environment, helping readers unfamiliar with [Agar.io](http://agar.io/) quickly grasp the core mechanics and objectives.
3. The paper is clearly written, visually well-organized, and supported by detailed figures and appendices that make the environment’s design, components, and experiments easy to follow.
4. The authors conduct numerous experiments across different settings and algorithms, providing a thorough empirical assessment and insightful analyses of learning behavior.
5. The introduction of mini-games is a useful contribution. These tasks serve as controlled, diagnostic environments that are useful not only for AgarCL but also for studying general RL behavior and subproblem difficulty.

**Weaknesses:**

1. **Continual?** The environment is not truly *continual* in the RL sense. In continual RL, the world itself changes while the agent’s policy persists: new objectives emerge, opponent distributions evolve, and goals shift. In AgarCL, the apparent “change” stems entirely from the agent’s own state: as mass increases, movement slows and the field of view expands, altering the interaction dynamics. These effects are endogenous and fully captured by a single stationary MDP, while the transition and observation functions remain fixed. The game is rather *continuing* (non-episodic) but not *continual*: the underlying rules never drift independently of the agent. Consequently, the collapse of frozen policies reflects poor generalization or behavioral distribution shift, not genuine environmental non-stationarity. Likewise, while it is clear that standard RL baselines lack CL capabilities, their low performance only indicates that AgarCL is a complex non-stationary environment. For AgarCL to be a continual RL environment, it would need exogenous drift, e.g., evolving opponent strategies, changing spawn rates, decaying resource yields, or irreversible world modifications. Without such dynamics, the task remains a single stationary environment with long horizons and internal variability. Although, while a great environment for regular RL, it lacks the continual component.
2. **Framing**. The paper implies that environments with gradual, endogenous shifts are inherently more realistic or valuable than those with abrupt task changes. However, I find this claim unconvincing. The relevance of abrupt versus smooth change depends entirely on the application domain. Think of scenarios, such as a warehouse robot deployed in a new facility with unseen layouts. The agent faces sudden distribution shifts, since the warehouse does not gradually evolve into a different one. The robot’s existing policy may fail to generalize, yet retraining from scratch is impractical. It should thus adapt to the new layout without forgetting past ones. The authors’ own observation that few existing continual RL methods work “out of the box” in AgarCL because they are tailored to the sequential tasks setting indicates that this CL setting remains highly relevant.
3. **Baselines**. Despite positioning the work as a continual RL benchmark, no actual CL methods are evaluated. All reported results come from standard RL algorithms (DQN, PPO, SAC). As a result, the paper does not demonstrate whether the environment meaningfully distinguishes CL capabilities.
4. **Deterministic opponents**. AgarCL relies on deterministic, hand-crafted bots, which introduces the risk of overfitting and exploitation rather than genuine learning. RL agents are well known for discovering loopholes with fixed opponents [1]. Although the evaluated RL baselines fail to obtain meaningful performance, given that AgarCL’s opponents are rule-based, the agent may simply learn to exploit their patterns instead of acquiring generalizable strategies. This undermines the claim that performance improvements reflect continual adaptation.
5. **Metrics**. Many continual RL benchmarks evaluate conventional CL metrics such as transfer and forgetting [2, 3, 4, 5, 6]. In contrast, this paper reports only cumulative reward as the primary metric.
6. **Derivative Design**. Even if the work were positioned purely as a new RL environment rather than continual, its contribution would not be groundbreaking, as there are existing Agar.io-style implementations for RL [7, 8, 9]. While the mini-games are useful tasks themselves, the pixel observations provide a new layer of complexity, and the notable simulation speed-up can reduce the runtime burden, the core mechanics and dynamics remain largely unchanged.

### Minor points

1. Using smoothing would improve the readability of Figures 4, 5, 16, and 17.
2. The frame skip is set to 4 in the environment specifications. This should be mentioned only in the experiments section, since this is generally up to the user to define.

[1] Delfosse, Quentin, et al. "Deep Reinforcement Learning Agents are not even close to Human Intelligence." *arXiv preprint arXiv:2505.21731* (2025).

[2] Powers, Sam, et al. "Cora: Benchmarks, baselines, and metrics as a platform for continual reinforcement learning agents." *Conference on Lifelong Learning Agents*. PMLR, 2022.

[3] Tomilin, Tristan, et al. "Coom: A game benchmark for continual reinforcement learning." *Advances in Neural Information Processing Systems* 36 (2023): 67794-67832.

[4] Johnson, Erik C., et al. "L2explorer: A lifelong reinforcement learning assessment environment." *arXiv preprint arXiv:2203.07454* (2022).

[5] Wołczyk, Maciej, et al. "Continual world: A robotic benchmark for continual reinforcement learning." *Advances in Neural Information Processing Systems* 34 (2021): 28496-28510.

[6] Tomilin, Tristan, et al. "MEAL: A Benchmark for Continual Multi-Agent Reinforcement Learning." *arXiv preprint arXiv:2506.14990* (2025).

[7] Zhang, Ming, et al. "Gobigger: A scalable platform for cooperative-competitive multi-agent interactive simulation." *The Eleventh International Conference on Learning Representations*. 2023.

[8] Ansó, Nil, et al. "Deep reinforcement learning for pellet eating in agar. IO." *The 11th International Conference on Agents and Artificial Intelligence*. SciTePress, 2019.

[9] Wiehe, Anton Orell, et al. "Sampled policy gradient for learning to play the game Agar. io." *arXiv preprint arXiv:1809.05763* (2018).

**Questions:**

1. Can pretrained policies be used as opponents?
2. How does the agent perceive the area outside the outer wall of the environment with the pixel-based observations? Are the pixels outside the area padded with some value?
3. Why do the pixel observations have separate channels for in-game objects? Is it necessary to semantically separate the input rather than use the RGB channels that a human player would see?

---

> ### Author Response · Authors · 2025-11-18
>
> We thank the reviewer for their feedback. We were pleased to see that you liked AgarCL, both the environment itself, including the action space and gameplay mechanics, as well as our presentation (“paper is clearly written, visually well-organized, and supported by detailed figures and appendices that make the environment’s design, components, and experiments easy to follow”). Given this positive feedback, we were initially surprised by a rating of 2, and we hope to have a productive discussion about the concerns raised to demonstrate that the paper is deserving of publication at ICLR.
>
> > “Continual? The environment is not truly continual in the RL sense…”
>
> We truly appreciate you pointing out this nuance, as we have always been deliberate about our word choice (see footnote 7). However, we disagree with your analysis, and the reason is largely a matter of perspective. In our paper, we take an experiential approach, where the primary concern is the agent’s data stream. From the agent’s perspective, the world changes because it cannot capture every aspect of the environment, perceiving it as non-stationary. The partial observability of the game plays a significant role here—a POMDP may appear non-stationary to an agent without access to the environment’s true states.
>
> To elaborate further, the agent does not have access to the environment’s dynamics. Even when other agents are stationary, it is challenging for the agent to model them accurately: it would need to track other players, identify which ones reappear, and realize they are stationary. Explicitly altering environment dynamics is delicate, as abrupt changes could be introduced instead of the slow, endogenous drift that naturally occurs, which is exceedingly difficult for the agent to model.
>
>
> >  “Framing. The paper implies that environments with gradual, endogenous shifts are inherently more realistic or valuable than those with abrupt task changes.”
>
> We agree with the reviewer that settings with abrupt changes are relevant and have important use cases. It was not our intention to suggest that environments with slow changes are inherently more valuable, only that they are currently less common in the community, and therefore offer a larger gap in terms of evaluation platforms. While we did state that we consider them more realistic, we recognize this is a matter of perspective and will ensure these claims are better contextualized in the version we submit next week.
>
> > “Baselines. Despite positioning the work as a continual RL benchmark, no actual CL methods are evaluated.”
>
> Yes, we initially chose to focus on traditional RL algorithms because most of the continual RL methods either rely on an explicit notion of tasks or are designed to tackle things such as loss of plasticity. We do not have tasks in AgarCL, and we did not observe evidence of loss of plasticity in our experiments. We wanted to introduce a new evaluation platform for continual RL instead of having to benchmark every continual RL algorithm in the field, thus focusing on the problem, not solution methods.
>
> In any case, note that we did evaluate continual backprop (see Appendix L.2) and we failed to see any gains. We are currently exploring additional boundary-agnostic CL approaches, such as Shrink-and-Perturb and ReDo, to evaluate further whether recent methods can help in AgarCL, and we will send an update alongside the new version of the paper.
>
> > "Deterministic opponents. AgarCL relies on deterministic, hand-crafted bots, which introduces the risk of overfitting and exploitation rather than genuine learning."
>
> Yes, this is an interesting trade-off we faced. Introducing opponents that also learn would reduce the environment’s framerate by orders of magnitude. We had to make a choice, and this was the compromise we selected. Note that we are free to choose among other fixed policies when an opponent respawns. We still believe this is a valuable intermediate step that allows us to explore many meaningful questions in continual RL, and the agent’s difficulty in succeeding in AgarCL supports this claim.
>
> Finally, note that GoBigger is an excellent platform for focusing on the multi-agent aspect of agar.io, but it does not emphasize continual learning.

---

> > ### Author Response · Authors · 2025-11-18
> >
> > > “Metrics. Many continual RL benchmarks evaluate conventional CL metrics such as transfer and forgetting [2, 3, 4, 5, 6]. In contrast, this paper reports only cumulative reward as the primary metric.”
> >
> > Yes, that is correct, and it was done intentionally. The total reward accumulated by the agent is the primary metric that any reinforcement learning agent ultimately cares about. Issues such as loss of plasticity or catastrophic forgetting naturally hinder reward accumulation. While evaluating these phenomena explicitly would be interesting, we believe that more effective algorithms capable of succeeding in AgarCL are needed first. For instance, we evaluated continual backpropagation in AgarCL (Appendix L), which still failed to perform well regardless of its ability to maintain plasticity.
> >
> > > “Derivative Design. Even if the work were positioned purely as a new RL environment rather than continual, its contribution would not be groundbreaking, as there are existing Agar.io-style implementations for RL [7, 8, 9]. While the mini-games are useful tasks themselves, the pixel observations provide a new layer of complexity, and the notable simulation speed-up can reduce the runtime burden, the core mechanics and dynamics remain largely unchanged.”
> >
> > Many of the environments that have shaped reinforcement learning research were themselves adaptations of existing games. Notable examples include Atari 2600 games, Minecraft, and StarCraft II. This approach is common for good reasons: such environments are free of experimenter bias, interpretable by humans, and come with clear performance metrics. Our contribution operationalizes this environment for RL research, with a focus on continual RL. While other papers have used variants of agar.io for evaluation, they did so without mentioning continual learning, which further underscores our novelty claim. In many ways, this is similar to what the Arcade Learning Environment (ALE) did with Atari 2600 games, leveraging existing games to highlight new challenges, such as generality in AI. We do not think most would consider ALE’s design “derivative,” even though the core mechanics of Atari 2600 games were unchanged and earlier work (Diuk et al., 2008) had already used Atari 2600 games for RL evaluation.

---

> > > ### Comment · Reviewer_Vnh3 · 2025-11-28
> > >
> > > >the reason is largely a matter of perspective
> > >
> > > Positioning your work as a continual RL benchmark inevitably sets certain expectations. If you reinterpret “*continual*” through a new lens, you might end up just muddying the water. Of course, this can be done [1], but it requires far more groundwork. I also don’t think it’s productive to get dragged into debating what continual RL is. A more practical viewpoint is to look at what continual RL researchers focus on. What are the challenges? What metrics are measured? What phenomena are investigated? Once that is established, we can think what kind of simulation environments help us to evaluate/solve/investigate these problems. I don’t, unfortunately, see this line of thinking in your paper to justify AgarCL as a meaningful tool for continual RL.
> > >
> > > >From the agent’s perspective, the world changes because it cannot capture every aspect of the environment, perceiving it as non-stationary. The partial observability of the game plays a significant role here—a POMDP may appear non-stationary to an agent without access to the environment’s true states.
> > >
> > > You are describing exactly what a non-stationary environment looks like. I don’t see where the *continual* aspect emerges. The smooth changes due to mass, speed, FOV, decay etc., are all functions of the current state inside a single stationary transition kernel. You even formalize this POMDP with a fixed transition function. Nothing in the world changes over time unless the agent grows or the built-in stochastic processes unfold (pellet spawn, virus spawn), but those spawn processes are stationary, with fixed rates.
> > >
> > > >Yes, we initially chose to focus on traditional RL algorithms
> > >
> > > Initially? This makes it sound like the paper is incomplete, and we should be expecting another iteration of baselines.
> > >
> > > >most of the continual RL methods either rely on an explicit notion of tasks or are designed to tackle things such as loss of plasticity
> > >
> > > This exactly reinforces my earlier point. You struggled to find CL baselines that fit your setting. I don’t think this is a gap in CL literature, but rather it suggests that the setting you propose has not been interesting/relevant for the CL community to shift their focus towards. If there are no CL methods from the past that are applicable to your environment, then what suggests that there would be in the future?
> > >
> > > Note that many CL methods don’t necessarily need task boundaries. You could make slight adjustments and still be able to run baselines with the same principles. To bring a few examples: from the reg-based methods family, you could apply a simple L2 regularization to the learned weights, or do something a bit smarter by periodically detecting important weights with importance sampling of MAS/EWC and apply the regularization term during policy updates. Likewise, you could adapt memory-based methods, such as AGEM/CLEAR, to periodically store important trajectories in a separate buffer and use that data at later stages of training.
> > >
> > > [1] Abel, David, et al. "A definition of continual reinforcement learning." *Advances in Neural Information Processing Systems* 36 (2023): 50377-50407.

---

> > > > ### Comment · Reviewer_Vnh3 · 2025-11-28
> > > >
> > > > >We wanted to introduce a new evaluation platform for continual RL instead of having to benchmark every continual RL algorithm in the field, thus focusing on the problem, not solution methods.
> > > > If the goal is to introduce an evaluation platform for continual RL, then what exactly are you evaluating that demonstrates continual learning? How do we know that the cumulative reward shows whether an agent is learning continually? In my view, it only shows how it performs in a non-stationary environment. I fail to se any metrics or analysis that distinguishes continual learning from plain adaptation in a stationary POMDP.
> > > >
> > > > Also, no one is expecting you to run every algorithm. The core issue that other reviewers and I have pointed out is that we don’t even see a single sensible regularization-based/rehearsal-based/architecture-based CL method evaluated.
> > > >
> > > > >In any case, note that we did evaluate continual backprop
> > > >
> > > > Continual backprop is a remedy for the loss of plasticity caused by curvature explosion. Since you state that you did not observe evidence of loss of plasticity in your experiments, then it is with little surprise that you also didn’t see any improvement gains. Therefore, I don’t see the motivation for evaluating continual backprop. The same goes for ReDo.
> > > >
> > > > > Issues such as loss of plasticity or catastrophic forgetting naturally hinder reward accumulation.
> > > >
> > > > While that might be true in many cases, it doesn’t justify omitting them. The evaluation is less informative.
> > > >
> > > > > While evaluating these phenomena explicitly would be interesting, we believe that more effective algorithms capable of succeeding in AgarCL are needed first
> > > >
> > > > The phrasing again implies as if we should be expecting an AgarCLv2, which finally incorporates many of the features you discuss. I believe these features ought to belong in the current version, not a hypothetical future update.
> > > >
> > > > >Many of the environments that have shaped reinforcement learning research were themselves adaptations of existing games.
> > > >
> > > > I fear that you misunderstood my point. I didn’t mean to say your work is derivative because it builds on the Agar.io game. My point was that it is derivative from existing simulation environments targeting Agar.io, many of which implement the same core mechanics and challenges. The fact that you use the word “*continual*” in the paper does not add much novelty in itself.
> > > >
> > > > >Given this positive feedback, we were initially surprised by a rating of 2.
> > > >
> > > > I do think the authors have made some useful and interesting additions to agar.io for RL. Also, the presentation and clarity of the work are very good. But I really don’t think the work (in its current state) should be positioned as a continual RL benchmark. Unfortunately, I am inclined to keep my score.

---

> > > > ### Author Response · Authors · 2025-11-28
> > > >
> > > > >You are describing exactly what a non-stationary environment looks like. I don’t see where the continual aspect emerges. (...) You even formalize this POMDP with a fixed transition function. Nothing in the world changes over time unless the agent grows or the built-in stochastic processes unfold (pellet spawn, virus spawn), but those spawn processes are stationary, with fixed rates.
> > > >
> > > > This point falls into exactly the trap you cautioned against earlier: debating what “continual RL” is. We maintain that a key aspect of continual learning is non-stationarity—and that is precisely what AgarCL presents. Formalizing the environment as a POMDP is irrelevant for this discussion: the agent cannot model the POMDP fully, and from the agent’s perspective, the world **is not** stationary. In essence, saying the world is stationary is equivalent to claiming that if one could model every atom in the universe, the distribution of the next state in the real-world could be perfectly predicted. What matters in continual learning is the agent’s experience of non-stationarity, not an omniscient formalization.
> > > >
> > > > > Initially? This makes it sound like the paper is incomplete, and we should be expecting another iteration of baselines.
> > > >
> > > > We used “initially” with the broader research agenda in mind, not to suggest that the paper is incomplete. We believe the most important contribution of this work is to characterize the behavior of traditional algorithms in AgarCL, rather than to provide a comprehensive table of methods primarily developed to address loss of plasticity or catastrophic forgetting—important topics, but not the only ones relevant to continual RL. Other fundamental questions, such as exploration and generalization, are central in this environment, and it is not clear that many existing methods meaningfully address them.
> > > >
> > > > > This exactly reinforces my earlier point. You struggled to find CL baselines that fit your setting. I don’t think this is a gap in CL literature, but rather it suggests that the setting you propose has not been interesting/relevant for the CL community to shift their focus towards. If there are no CL methods from the past that are applicable to your environment, then what suggests that there would be in the future?
> > > >
> > > > Many times, a new problem serves as a catalyst for the creation of new approaches. It would be like claiming that ImageNet had no value because it was hard to imagine a classification algorithm capable of processing that many images, or that Atari 2600 games had no value because it was hard to imagine an approach that could obtain reasonable representations across 50 different games.
> > > >
> > > > > Continual backprop is a remedy for the loss of plasticity caused by curvature explosion. Since you state that you did not observe evidence of loss of plasticity in your experiments, then it is with little surprise that you also didn’t see any improvement gains. Therefore, I don’t see the motivation for evaluating continual backprop. The same goes for ReDo.
> > > >
> > > > Much of the current literature focuses on loss of plasticity. We chose Continual Backprop because it is a recent, high-profile algorithm. But in our platform, we do not see evidence of catastrophic forgetting, so the recommendation to evaluate CLEAR is not clearly motivated. Similarly, we do not see overfitting, so the suggestion to test a regularization-based method lacks a clear rationale. Ultimately, this can devolve into a cycle of complaints because a reviewer’s preferred algorithm was not evaluated.
> > > >
> > > > We reordered our comments to address this point early, which also justifies why we continue to emphasize that Continual Backprop is evaluated throughout the response.
> > > >
> > > > > Note that many CL methods don’t necessarily need task boundaries. You could make slight adjustments and still be able to run baselines with the same principles. To bring a few examples: from the reg-based methods family, you could apply a simple L2 regularization to the learned weights, or do something a bit smarter (...) you could adapt memory-based methods, such as AGEM/CLEAR, to periodically store important trajectories in a separate buffer...
> > > >
> > > > Admittedly, we missed CLEAR, although the notion of “important trajectories” would need to be explicitly defined. Note that our original submission already includes an evaluation of continual backpropagation (Appendix L.2). We are currently finalizing experiments with two additional boundary-agnostic methods—ReDo and Shrink-and-Perturb—and will include these results in the revised version of the paper. Unfortunately, the short window between the reviewer response and the end of the discussion period does not allow us to reliably evaluate additional algorithms, especially given the potentially unbounded list of adaptations one could consider. We maintain that evaluating more baselines is not the main contribution of this paper.

---

> ### Author Response · Authors · 2025-11-28
>
> > Also, no one is expecting you to run every algorithm. The core issue that other reviewers and I have pointed out is that we don’t even see a single sensible regularization-based/rehearsal-based/architecture-based CL method evaluated.
>
> As we pointed out, we did evaluate Continual Backprop. We genuinely struggle to see what one would gain from this evaluation. For example, do we really think that a regularization-based method will magically solve the issues the agents faced regarding exploration without resets? The contributions of the paper are to introduce a new platform for research in which a fixed policy performs worse than one that is continually learned.
>
> We demonstrate that there are interesting challenges to be addressed, and we isolate many of them through mini-games.
> The question is not whether we could have done more, one always can, but whether what we are presenting contributes to the field. Unfortunately, the pattern of the field is to always ask for more. If we had evaluated “a single sensible regularization-based/rehearsal-based/architecture-based CL method”, reviewers would complain about the fact we did not evaluate more (we did evaluate Continual Backprop afterall). If we evaluated more, reviewers would complain we didn’t consider recurrent architectures given the partial observability of the problem (which we did in our paper). And so on and so forth.
>
> > The phrasing again implies as if we should be expecting an AgarCLv2, which finally incorporates many of the features you discuss. I believe these features ought to belong in the current version, not a hypothetical future update.
>
> This interpretation injects intent where none exists and comes across as somewhat cynical. We have been explicit throughout the review process that this paper is about presenting an evaluation platform, and we stand by that focus. The future is not a hypothetical AgarCLv2; rather, it consists of other papers that will build on top of AgarCL.
>
> > I fear that you misunderstood my point. I didn’t mean to say your work is derivative because it builds on the Agar.io game. My point was that it is derivative from existing simulation environments targeting Agar.io, many of which implement the same core mechanics and challenges. The fact that you use the word “continual” in the paper does not add much novelty in itself.
>
> Others have not released an evaluation platform that can serve as a foundation for further research; GoBigger, which has a fundamentally different focus, is the only exception. Our contribution is not simply using the word “continual”—it is the design of an environment that genuinely supports continual learning research. For example, prior work typically consists of short episodes that frequently reset, whereas we designed a reward function and interaction mechanism that enable continual learning. We also went out of our way in designing the mini-games to isolate and highlight many key aspects of the environment.

---

### Official Review · Reviewer_CrrQ · 2025-10-29

**Soundness:** 3
**Presentation:** 3
**Contribution:** 2
**Rating:** 4
**Confidence:** 4

**Summary:**

The paper introduces AgarCL, a new benchmark environment for continual reinforcement learning (RL) based on the game Agar.io. Unlike traditional episodic RL tasks, AgarCL is non-episodic, partially observable, and non-stationary, requiring agents to adapt continuously as the environment evolves.
AgarCL features hybrid actions (continuous movement + discrete split/eject), pixel-based observations, and mass-based rewards, creating smooth but persistent changes in dynamics as agents grow or shrink. The authors benchmark DQN, PPO, and SAC, showing that none can learn stable or effective policies in the full environment.
They also design mini-games isolating specific challenges (exploration, credit assignment, non-stationarity) and show PPO performs best but still struggles. Fixed policies degrade over time, demonstrating the need for continual adaptation.

**Strengths:**

The paper presents a clear and well-structured formalization of the problem, with solid motivation and coherent methodology. The presentation is generally good, and the theoretical framing is appealing

**Weaknesses:**

The contribution lacks strong novelty, as it mostly adapts an existing game setup rather than introducing new concepts. The analysis of opponent policies could be expanded, for example by addressing non-stationary behaviors. While the focus is not on continual learning (CL), it would be valuable to highlight the method’s compatibility with existing CL frameworks. Finally, a few figures and tables would benefit from clearer legends for better readability.

**Questions:**

While the work is not explicitly about continual learning, do you plan to make AgarCL compatible with existing continual learning frameworks or benchmarks (e.g., through defined task boundaries, curriculum setups, or standardized evaluation metrics)?

Do you plan to include or analyze non-stationary opponent behaviors to better reflect continual adaptation challenges and multi-agent dynamics?

---

> ### Author Response · Authors · 2025-11-18
>
> We thank the reviewer for their thoughtful and constructive feedback. We are glad that you found the problem formulation clear, the motivation well grounded, and the overall presentation strong. Given this positive assessment and the fact that the concerns you raised can be readily addressed, we hope the discussion below will help improve the rating of the paper. If not, we would appreciate clarification, as we believe the issues should be straightforward to resolve.
>
> > "The contribution lacks strong novelty, as it mostly adapts an existing game setup rather than introducing new concepts. The analysis of opponent policies could be expanded, for example by addressing non-stationary behaviors. While the focus is not on continual learning (CL), it would be valuable to highlight the method’s compatibility with existing CL frameworks. Finally, a few figures and tables would benefit from clearer legends for better readability."
>
> Let us start by pointing out that many of the environments that have shaped reinforcement learning research were themselves adaptations of existing games. Notable examples include Atari 2600 games, Minecraft, and StarCraft II. This approach is common for good reasons: such environments are free of experimenter bias, interpretable by humans, and come with clear performance metrics. Our contribution operationalizes this environment for RL research, with a focus on continual RL. Even other papers that have used variants of agar.io for evaluation did so without any mention of continual learning, further underscoring our novelty claim. In many ways, this is similar to what the Arcade Learning Environment did with Atari 2600 games, leveraging existing games to highlight new challenges such as generality in AI.
>
> Additionally, we discuss in Appendix L.2 the performance of an existing continual learning solution in AgarCL, Continual Backprop, and explicitly discuss the limitations of other solution methods in the same section. We also discuss other continual learning platforms throughout the main paper, particularly in Section 5. Would your concern be addressed if we moved the discussion from Appendix L.2 into the main paper, as we have promised to other reviewers?
>
> Finally, could you clarify what you mean by expanding the “analysis of opponent policies,” and indicate which figures and tables you feel would “benefit from clearer legends”?
>
> > “While the work is not explicitly about continual learning, do you plan to make AgarCL compatible with existing continual learning frameworks or benchmarks (e.g., through defined task boundaries, curriculum setups, or standardized evaluation metrics)?”
>
> We will not introduce task boundaries to AgarCL, as doing so would defeat the core motivation of the platform. The idea of using a sequence of mini-games as a curriculum is an interesting research direction that we plan to explore in the future, but it is beyond the scope of this paper, which focuses on introducing the research platform.
>
> Regarding research metrics, the total amount of reward accumulated by the agent is the central metric that any reinforcement learning agent ultimately cares about. If issues such as loss of plasticity or catastrophic forgetting arise, they will directly hinder the agent’s ability to accumulate reward. We agree that explicitly evaluating these phenomena would be interesting, but we believe that better algorithms capable of succeeding in AgarCL are needed before such metrics become informative. For instance, we did evaluate continual backpropagation in AgarCL (Appendix L), but it also failed to perform well in the environment, regardless of its ability to maintain plasticity.
>
> AgarCL is intentionally boundary-free, reflecting settings where the agent must continually adapt without discrete task labels. Many continual learning methods—such as EWC—assume explicit task boundaries and are therefore incompatible with this setting by design. However, we have already tested a boundary-agnostic CL method (continual backpropagation) and are currently exploring additional approaches, such as Shrink-and-Perturb and ReDo, to further assess whether recent methods can succeed in AgarCL.

---

> > ### Author Response · Authors · 2025-11-18
> >
> > >“Do you plan to include or analyze non-stationary opponent behaviors?”
> >
> > We agree that analyzing non-stationary opponent behaviors is a very interesting research question, but it is outside the scope of this paper. We do plan to add such a feature in the future, but it would be disingenuous to make any promises about it during the rebuttal phase, as this would constitute a separate paper.
> >
> > Nevertheless, even fixed opponents induce non-stationarity because:
> >
> > * The agent cannot distinguish opponent types from visual appearance alone (shared colors, partial observability).
> > * Different opponent strategies produce varying local dynamics and threat levels.
> > * The zoom-out mechanism changes pixel density as the agent grows, introducing additional perceptual non-stationarity and a functional change in the reward structure.
> >
> > Thus, even with fixed policies, the agent faces uncertainty and shifting interaction patterns that are functionally similar to dealing with non-stationary opponents.

---

> ### Comment · Reviewer_CrrQ · 2025-11-27
> **Final Score**
>
> Thank you for the detailed responses and clarifications. I appreciate the authors’ efforts to address the points I raised,particularly regarding the formalization of the environment, the discussion of continual learning methods, and the broader positioning relative to prior game-based RL benchmarks.
>
> However, while these revisions improve the clarity of the submission, they do not substantially change my overall assessment of the novelty and contribution. My main concerns remain:
>
> Continual learning novelty: The work frames AgarCL as a continual RL benchmark, but the continual-learning aspects rely primarily on properties inherent to the Agar.io dynamics rather than introducing new continual-learning concepts, mechanisms, or evaluation principles. I would have expected a clearer contribution toward continual learning,whether through tailored metrics, structured evaluation protocols, or integration with established CL paradigms,rather than a derivative adaptation of an existing game environment.
>
> Opponent analysis and non-stationarity: The analysis of opponents, especially in relation to non-stationary behavior, remains limited. For a benchmark positioned around continual adaptation, deeper examination of opponent dynamics, opponent-induced non-stationarity, or adaptive multi-agent interactions would meaningfully strengthen the contribution.
>
> Derivative design: While many RL environments originate from existing games, the current adaptation does not yet introduce enough methodological or conceptual novelty to justify a higher evaluation. The environment is interesting and potentially useful, but the contribution remains primarily an engineering adaptation rather than an advance in continual RL research.
>
> Given these persistent concerns, I will maintain my original score.

---

> > ### Author Response · Authors · 2025-11-28
> >
> > Thank you for responding during the rebuttal period. It appears there is a chasm between our view of what constitutes a valid contribution and the reviewer’s expectations. Our response below aims to clarify this gap.
> >
> > > Continual learning novelty: The work frames AgarCL as a continual RL benchmark, but the continual-learning aspects rely primarily on properties inherent to the Agar.io dynamics rather than introducing new continual-learning concepts, mechanisms, or evaluation principles.
> >
> > This is exactly the point: our goal is to operationalize continual RL research in a platform we believe is well-suited for it. The mini-games clearly isolate components of the game that create non-stationarity, and adapting Agar.io to RL required numerous design choices that fundamentally shape the platform, including a distinct mechanism for introducing stochasticity (via action noise), hybrid actions, our modeling of death and reward, and the use of fixed agents drawn from diverse policies. Several prior agar.io-like platforms relied on very different instantiations that do not elicit continual learning.
> >
> > So, again, we fundamentally disagree with the claim of “derivative design.”
> >
> > > (...) but the contribution remains primarily an engineering adaptation rather than an advance in continual RL research.
> >
> > The contribution of AgarCL is precisely to enable future advances in continual RL research. Similarly, when the ALE was introduced, the contribution was not in immediately advancing research on general agents, but in providing a platform that made such advances possible. The same can be said about Minecraft as a research platform for RL.
> >
> > >  Opponent analysis and non-stationarity: The analysis of opponents, especially in relation to non-stationary behavior, remains limited. For a benchmark positioned around continual adaptation, deeper examination of opponent dynamics, opponent-induced non-stationarity, or adaptive multi-agent interactions would meaningfully strengthen the contribution.
> >
> > One of the motivating perspectives in continual learning is that the environment is always bigger than the agent—often because it contains other agents of comparable capability. For this reason, we disagree with the idea that multi-agent influences must be disentangled from continual RL; in many settings, these factors are fundamentally intertwined. We do not frame AgarCL as a MARL problem because our focus is explicitly on the experiential perspective of the learning agent, treating the behavior of other entities as part of the environment’s dynamics.
> >
> > Expecting a new benchmark to both introduce the environment and fully resolve all questions about multi-agent structure within a single paper sets an unreasonable bar. Our goal in this work is to provide the platform that enables such investigations—not to preemptively answer all of them.

---

### Official Review · Reviewer_djmq · 2025-10-30

**Soundness:** 2
**Presentation:** 2
**Contribution:** 2
**Rating:** 4
**Confidence:** 4

**Summary:**

The manuscript introduces AgarCL, a continual reinforcement learning research platform derived from the Agar.io game. The environment is non-episodic, high-dimensional, partially observable, and features continuous actions and endogenous non-stationarity due to evolving dynamics and other agents. The authors position AgarCL as a testbed that avoids artificial task switches commonly used to induce non-stationarity in episodic benchmarks. They provide baseline results for DQN, PPO, and SAC on both the full environment and a suite of mini-games designed to isolate specific challenges. The results suggest that fixed policies struggle to maintain stable performance and that standard deep RL methods face considerable difficulty in this setting. The manuscript emphasizes problem formulation and environment contribution over new algorithmic solutions, and acknowledges substantial computational demands

**Strengths:**

1. The manuscript targets a gap in CRL benchmarks by providing a non-episodic environment with smooth endogenous non-stationarity, moving beyond artificial task switches common in prior work.
2. The environment’s combination of partial observability, continuous control, high-dimensional observations, and potentially infinite horizon is well aligned with realistic continual learning challenges.
3. The manuscript provides a detailed description of the platform and its dynamics, which can help researchers understand and instrument experiments in this setting.
4. Due to well-engineered and accessible, AgarCL may serve as a common platform that encourages standardized evaluation of RL approaches in non-episodic settings.

**Weaknesses:**

1. The originality of the platform is limited, as it largely adapts Agar.io for RL without clear evidence of novel environment design beyond configuration.
2. The evaluation does not include algorithms specifically designed for CRL (or methods targeting non-stationarity, online adaptation, memory consolidation, or meta-learning), making it hard to assess whether the environment differentiates among approaches intended for this setting.
3. The baselines (DQN, PPO, SAC) are standard and not state-of-the-art for the reported setup; tuning procedures and fairness across methods are insufficiently detailed, and the results provide limited actionable guidance for algorithm development.
4. The manuscript’s positioning of AgarCL as a CRL benchmark is blurred by extensive experiments in non-continual configurations and by the strong influence of other agents; it risks being better framed as a multi-agent platform without providing corresponding multi-agent protocols or analyses.
5. The structure reads more like a technical report than a concise research manuscript; core contributions and key takeaways are not sharply distilled, and many content appears relegated to appendices without synthesis in the main text.
6. The evaluation protocol lacks standard continual learning metrics (e.g., forgetting, forward/backward transfer, stability–plasticity trade-offs) and does not establish clear, reproducible benchmarks for long-horizon continual performance.
7. Practical considerations (compute requirements, scalability, parallelization, runtime) are acknowledged but not resolved; heavy resource demands limit accessibility and may impede community adoption.
8. Although the authors deserve credit for their work in developing this platform, the manuscript does not contribute new knowledge and sufficient value to the ICLR community. This is perhaps the wrong venue for this work. The contribution is primarily infrastructural and may be better suited to a specialized track.

**Questions:**

Please refer to the weaknesses part.

---

> ### Author Response · Authors · 2025-11-18
>
> We thank the reviewer for their feedback. We were pleased to see that you also find the proposed environment well aligned with realistic continual learning challenges and recognize its potential as a common platform for standardized evaluation of RL approaches in non-episodic settings. We believe these points directly support the case for acceptance, and we hope that our responses below clarify that the concerns raised do not warrant rejection.
>
> > “1. The originality of the platform is limited, as it largely adapts Agar.io for RL without clear evidence of novel environment design beyond configuration.”
>
> Many environments that have shaped reinforcement learning research were themselves adaptations of existing games. Notable examples include Atari 2600 games, Minecraft, and StarCraft II. This approach is common for good reasons: such environments are free of experimenter bias, interpretable to humans, and come with clear performance metrics. Our contribution is to operationalize this particular environment for RL research with a focus on continual RL. While other papers have used variants of agar.io for evaluation, they did so without any connection to continual learning, which further underscores our novelty claim. In this sense, our work parallels what the Arcade Learning Environment (ALE) did with Atari 2600 games, leveraging existing gameplay to highlight new challenges, such as generality in AI.
>
> > “2. The evaluation does not include algorithms specifically designed for CRL (or methods targeting non-stationarity, online adaptation, memory consolidation, or meta-learning), making it hard to assess whether the environment differentiates among approaches intended for this setting.”
>
> Yes, we initially chose to focus on traditional RL algorithms because most continual RL methods either rely on an explicit notion of tasks or are designed to address issues like loss of plasticity. AgarCL does not include tasks, and we did not observe evidence of loss of plasticity in our experiments. Our goal was to introduce a new evaluation platform for continual RL, rather than benchmark every existing continual RL algorithm; in other words, we aimed to highlight the problem setting rather than specific solution methods.
>
> In any case, we did evaluate continual backprop (see Appendix L.2) and did not observe any improvements. We are currently exploring additional boundary-agnostic CL approaches, such as Shrink-and-Perturb and ReDo, to further assess whether recent methods help in AgarCL, and we will provide an update alongside the revised version of the paper.
>
> > “3. The baselines (DQN, PPO, SAC) are standard and not state-of-the-art for the reported setup; tuning procedures and fairness across methods are insufficiently detailed, and the results provide limited actionable guidance for algorithm development.”
>
> We evaluated DQN, PPO, and SAC precisely because they are the standard approaches used when tackling a new RL problem. We are unsure what the criticism regarding tuning procedures and fairness refers to, as these details are covered extensively across multiple sections of the Appendix (see, for example, Appendices E, G, and H), spanning more than five pages. It is also unclear what the reviewer means by “results provide limited actionable guidance for algorithm development.”
>
> > “4. The manuscript’s positioning of AgarCL as a CRL benchmark is blurred by extensive experiments in non-continual configurations and by the strong influence of other agents; it risks being better framed as a multi-agent platform without providing corresponding multi-agent protocols or analyses.”
>
> A fundamental aspect of a continual RL benchmark is that the agent experiences non-stationarity, and, in AgarCL, part of that non-stationarity naturally comes from other agents. Importantly, the agents we include have fixed policies, and we clearly state in the paper that GoBigger is the appropriate platform for MARL investigations.
> Regarding the experiments, we believe that including non-continual configurations is necessary to quantify the effect of the continual setup; without this comparison, that question would remain unanswered. We also appreciate that the reviewer notes the extent of our experiments, as we conduct the same level of analysis for both continual and non-continual configurations.

---

> > ### Author Response · Authors · 2025-11-18
> >
> > > “5. The structure reads more like a technical report than a concise research manuscript; core contributions and key takeaways are not sharply distilled, and many content appears relegated to appendices without synthesis in the main text.”
> >
> > All other reviewers praised the clarity and overall presentation of our manuscript, so we do not understand why it would be characterized as a “technical report.” We would appreciate clarification on what the reviewer believes is fundamentally wrong with the structure. As noted in our responses to other reviewers, the revised version of the paper provides an extra page of content, which we will use to bring additional results from the appendix into the main text. Nonetheless, the consensus among the other reviews is that we describe our contributions and key takeaways clearly and, given the space constraints, focus on the empirical results that matter most.
> >
> > > “6. The evaluation protocol lacks standard continual learning metrics (e.g., forgetting, forward/backward transfer, stability–plasticity trade-offs) and does not establish clear, reproducible benchmarks for long-horizon continual performance.”
> >
> > Yes, that is true, and it was intentional. The total amount of reward accumulated by the agent is the central metric that any reinforcement learning agent ultimately cares about. If issues such as loss of plasticity or catastrophic forgetting arise, they will directly hinder the agent’s ability to accumulate reward. We agree that explicitly evaluating these phenomena would be interesting, but we believe that better algorithms capable of succeeding in AgarCL are needed before such metrics become informative. For instance, we did evaluate continual backpropagation in AgarCL (Appendix L), but it also failed to perform well in the environment, regardless of its ability to maintain plasticity.
> >
> > > “7. Practical considerations (compute requirements, scalability, parallelization, runtime) are acknowledged but not resolved; heavy resource demands limit accessibility and may impede community adoption.”
> >
> > We disagree that not fully resolving the compute requirements is a key weakness of the work. We have already improved simulation speed by orders of magnitude, but beyond that there are inherent limits to what can be optimized. Compute availability roughly doubles every two years, so running AgarCL will only become cheaper over time. Moreover, many established environments in the field, such as Minecraft and Atari 2600 games, have historically been heavy to run, yet this has not impeded community adoption. Even today, there is ongoing research on making those platforms more efficient through parallelism or alternative evaluation methods. We therefore think it is unfair to judge our platform harshly on this basis; if the same standard had been applied when many other influential environments were introduced, they might not have ever been published.
> >
> > > “8. Although the authors deserve credit for their work in developing this platform, the manuscript does not contribute new knowledge and sufficient value to the ICLR community. This is perhaps the wrong venue for this work. The contribution is primarily infrastructural and may be better suited to a specialized track.”
> >
> > We fundamentally disagree with this claim and look forward to engaging with the reviewer during this discussion phase. In short, we have introduced a new evaluation platform for continual learning research that exposes limitations of existing approaches, enables new lines of research, and provides a series of intermediate environments to accelerate experimentation and prototyping. All other reviewers have recognized the value of the proposed platform, and we hope our responses have addressed this concern.

---

> > > ### Comment · Reviewer_djmq · 2025-11-26
> > > **Response of "The Cell Must Go On: Agar.io for Continual Reinforcement Learning"**
> > >
> > > Thank you for the detailed rebuttal. While I appreciate the clarifications, several of the core concerns remain insufficiently addressed.
> > >
> > > W1
> > >
> > > It is not the use of an existing game per se that limits originality, but the absence of novel environment design choices that advance CRL evaluation. Historical precedents such as ALE did more than wrap Atari 2600 games: they introduced critical, widely adopted design elements (e.g., state save/restore mechanisms, frame skipping), and later refinements like sticky actions that shaped evaluation protocols and exposed meaningful algorithmic weaknesses [1, 2]. In contrast, the manuscript does not yet articulate comparably novel design decisions tailored to CRL. Given the availability of multiple CRL benchmarks today, merely operationalizing Agar.io without clear novel design features is unlikely to meet the expectations of ICLR 2026. Concretely, please identify and foreground environment-level innovations that are unique to AgarCL and demonstrably necessary for CRL evaluation, akin to how ALE’s design choices became integral to community practice.
> > >
> > > W2
> > >
> > > The claim that most CRL methods rely on explicit task boundaries overlooks a class of task-agnostic approaches that were explicitly designed not to depend on boundary signals. CLEAR is a prime example; the original paper emphasizes it does not require task boundaries [3]. Moreover, recent CRL benchmarks routinely include evaluations of CRL algorithms[4, 5]. To substantiate the manuscript’s value as a CRL benchmark, I strongly recommend adding results for task-agnostic methods. In particular, please evaluate CLEAR [3], 3RL [6], PT-TD [7], and Losse [8] under AgarCL’s continual setting. These additions would directly test whether AgarCL differentiates among approaches intended for boundary-agnostic continual adaptation. If certain methods are inapplicable, please provide precise justifications and, where feasible, adaptations consistent with prior art.
> > >
> > > W3
> > >
> > > A benchmark should yield actionable insights for algorithm developers, not only aggregate performance curves. For instance, CORA [5] provides: clearly defined metrics; guidance for integrating new algorithms; analyses of how design choices (e.g., CLEAR’s replay buffer size) affect outcomes; and concrete evidence about state-of-the-art limitations on long task sequences and sample efficiency. The current manuscript does not synthesize comparable guidance from its results, making it unclear how AgarCL helps drive CRL method development.
> > >
> > > W4
> > >
> > > Your own results (e.g., 4.3) indicate that a major driver of difficulty is the presence of many other agents. Because AgarCL’s non-stationarity arises from multiple sources, isolating these factors is essential to produce diagnostic experiments that advance CRL. The manuscript currently blurs CRL-specific claims with multi-agent influences, without providing protocols characteristic of multi-agent research.
> > >
> > > W5
> > >
> > > The concern is not a structural flaw but the lack of a crisp distillation of CRL-relevant contributions. For example, Chapter 3 elaborates Agar.io mechanics extensively, yet it remains unclear which elements constitute AgarCL’s novel design and how each ties to continual learning. In the experiments, substantial effort is devoted to non-continual configurations and broader analyses while key CRL baselines and metrics are largely absent, which dilutes the CRL message.
> > >
> > > W7
> > >
> > > While compute constraints need not be a fatal flaw, they do materially affect accessibility, especially when the platform’s
> > > novelty is modest. You note orders-of-magnitude speedups, but the manuscript does not detail the engineering or algorithmic changes that achieved them. If runtime improvements are a contribution of AgarCL, please document them explicitly. This transparency will help the community assess feasibility and adopt the platform more readily.
> > >
> > > W8
> > >
> > > As it stands, the platform does not convincingly expose limitations of existing CRL methods, offer a novel perspective, or provide substantial usability advances. If the revision can: (i) add thorough evaluations of CRL methods; (ii) incorporate standard continual metrics and ablations isolating non-stationarity sources; (iii) articulate environment design innovations; and (iv) document compute optimizations and integration guidance, I would be inclined to reconsider my assessment. Concentrating on CRL and delivering a comprehensive, reproducible continual evaluation protocol would make the work more compelling for the ICLR community.

---

> > > > ### Comment · Reviewer_djmq · 2025-11-26
> > > > **References**
> > > >
> > > > [1] Bellemare et al. “The Arcade Learning Environment: an Evaluation Platform for General Agents,” CoRR (2015).
> > > >
> > > > [2] Machado et al. “Revisiting the Arcade Learning Environment: Evaluation Protocols and Open Problems for General Agents,” CoRR (2018).
> > > >
> > > > [3] Rolnick et al. “Experience Replay for Continual Learning,” ICLR (2019).
> > > >
> > > > [4] Tomilin et al. “COOM: A Game Benchmark for Continual Reinforcement Learning,” NeurIPS (2023).
> > > >
> > > > [5] Powers et al. “CORA: Benchmarks, Baselines, and Metrics As a Platform for Continual Reinforcement Learning Agents,” CoRR (2022).
> > > >
> > > > [6] Caccia et al. “Task-Agnostic Continual Reinforcement Learning: Gaining Insights and Overcoming Challenges,” CoRR (2023).
> > > >
> > > > [7] Anand and Precup. “Prediction and Control in Continual Reinforcement Learning,” NeurIPS (2023).
> > > >
> > > > [8] Liu et al. “Locality Sensitive Sparse Encoding for Learning World Models Online,” CoRR (2024).

---

> > > > > ### Author Response · Authors · 2025-11-28
> > > > >
> > > > > Thank you for engaging with our rebuttal. Below, we clarify the points you raised and respond to several aspects where we fundamentally disagree.
> > > > >
> > > > > **W1**
> > > > >
> > > > > We fundamentally disagree with the reviewer’s assessment. It represents a revisionist view of platforms such as the ALE. When adapting an existing game into an evaluation platform, one inevitably makes design decisions that, if the platform is adopted by the community, may later become de facto standards. Ironically, many of the “features” the reviewer cites as essential were introduced years after the ALE was published, were later modified, or never became widely used.
> > > > >
> > > > > For example, sticky actions were introduced by Machado et al. (2018), five years after the original ALE paper—would we retroactively hold that against the original work? Frame skipping was originally set to 5 and only shifted to 4 after DQN became influential. Save/restore mechanisms were never broadly adopted. Importantly, none of these elements were articulated as contributions in the ALE’s initial introduction.
> > > > >
> > > > > AgarCL likewise includes deliberate design choices that depart from the original agar.io game, including frame skipping, a distinct mechanism for introducing stochasticity (via action noise), hybrid actions, our modelling of death and reward, and the use of fixed agents drawn from diverse policies. Several prior agar.io-like platforms used very different instantiations that do not elicit continual learning. And of course, the design of the mini-games is unique to our platform.
> > > > >
> > > > > Ultimately, AgarCL includes more than enough substantive design elements—just as every adapted game environment does—but many are low-level implementation details not appropriate to foreground as primary contributions. Our focus is on providing a coherent, well-motivated continual RL platform, not on enumerating every minor engineering choice.
> > > > >
> > > > > **W2**
> > > > >
> > > > > Thanks for pointing out CLEAR—this is indeed an interesting technique that does not rely on task boundaries, and we agree it deserves explicit acknowledgment in the paper. Notice, however, that the CLEAR paper introduces an *algorithm*, not an evaluation platform, and its experiments were conducted on Atari precisely because environments such as AgarCL did not exist at the time.
> > > > >
> > > > > Regarding continual RL baselines in AgarCL, our original submission already includes an evaluation of continual backpropagation (Appendix L.2). We are currently finalizing experiments with two additional boundary-agnostic methods—ReDo and Shrink-and-Perturb—and we will include these results in the revised version of the paper. Unfortunately, the short window between the reviewer response and the end of the discussion period does not give us enough time to reliably evaluate *four* additional algorithms.
> > > > >
> > > > > **W3**
> > > > >
> > > > > Again, we disagree. This is precisely why we introduced many of the mini-games discussed in the paper. Our results already highlight several core challenges in continual learning: the difficulty introduced by exploration, the limited benefits of recurrent architectures in this setting, the lack of performance gains from methods such as continual backpropagation, and the sensitivity of algorithms to factors such as mass decay. These findings directly expose important research gaps, and we have clearly outlined multiple promising directions for future work. Many of the challenges have already been documented in the current version of the paper.
> > > > >
> > > > > **W4**
> > > > >
> > > > > One of the motivating perspectives in continual learning is that the environment is always bigger than the agent—often because it contains other agents of comparable capability. For this reason, we disagree with the idea that multi-agent influences must be disentangled from continual RL; in many settings, these factors are fundamentally intertwined. We do not frame AgarCL as a MARL problem because our focus is explicitly on the experiential perspective of the learning agent, treating the behavior of other entities as part of the environment’s dynamics.
> > > > >
> > > > > Expecting a new benchmark to both introduce the environment and fully resolve all questions about multi-agent structure within a single paper sets an unreasonable bar. Our goal in this work is to provide the platform that enables such investigations—not to preemptively answer all of them.

---

> ### Author Response · Authors · 2025-11-28
>
> **W5**
>
> We believe our earlier response already justifies both the use of minigames and their role in addressing exactly the kind of analysis the reviewer requested in a separate comment. We deliberately chose not to frame the “novelty” of individual design elements as a primary contribution, because—much like the ALE and other influential platforms—the value of a new evaluation environment lies not in isolated mechanics but in the coherent research setting it enables. Focusing on low-level design differences misses the core purpose of introducing a platform: to provide a setting that exposes important challenges and catalyzes new research directions.
>
> **W7**
>
> This is not a technical report, contrary to what was suggested. The platform’s speed is a direct consequence of numerous design choices we made during implementation, from the programming language to the details of our collision-detection system. We explicitly report these performance characteristics: Appendix C discusses simulation speed, and Appendix F highlights the substantial performance difference relative to GoBigger. As stated there,
> > “Our environment significantly outperforms GoBigger in simulation speed, achieving an interquartile mean (IQM) of 4,212 fps with GoBigger-style observations, compared to GoBigger’s IQM of 205 fps under the same observation setup. This experiment was run using 2 cores of an Intel Xeon Gold 6448Y CPU.”
>
> **W8**
>
> We want to point out that many of the requests amount to writing a substantially different paper, one that would either span 50 pages or end up strictly worse than the current version. Nonetheless, we felt it important to address the points raised, as we believe the criticisms are not well founded.
>
> We did (i) evaluate a representative continual RL method in the submission and are adding additional ones during the rebuttal period, (ii) design minigames that explicitly isolate different sources of non-stationarity, (iii) keep minor implementation details out of the main paper because they are not the core contribution, and (iv) provide compute optimizations and integration guidance directly in the released source code and documentation. As the reviewer originally noted, ICLR is not aiming to encourage technical-report-style submissions, and our paper was written accordingly.

---

### Meta-Review · Area_Chair_p5RV · 2026-01-04

**Summary:**

The paper proposes AgarCL, a high-throughput (FPS) benchmark for Continual Reinforcement Learning (CRL) based on Agar.io. The authors posit that "experiential non-stationarity"—where the agent's irreversible growth fundamentally alters its interaction dynamics—provides a more natural testbed for CRL than artificial task-switching setups.

The decision to Reject is based on a fundamental flaw identified by the reviewers that remain unresolved after rebuttal: the authors' new experiments show that specialized CRL algorithms (Continual Backprop, ReDo), designed to mitigate forgetting and plasticity loss, fail to outperform standard RL baselines (PPO). This suggests that the specific form of non-stationarity in AgarCL does not induce the learning pathologies (e.g., gradient interference) that CRL methods are intended to solve. Instead, the challenge appears to be one of exploration or robustness within a hard-POMDP. Without demonstrating that "continual learning" is indeed the bottleneck, the paper cannot justify its positioning as a CRL benchmark.

**Reviewer Concerns:**

Addressed Concerns:
- Throughput & Engineering: The authors successfully clarified the significant speed advantage over GoBigger (4212 FPS vs 205 FPS), establishing the engineering value of the platform.
- Algorithm Inclusion: The authors added Continual Backprop, ReDo, and Shrink-and-Perturb to the evaluation.

Outstanding Concerns:
- Methodological Disconnect (Critical): As noted by Reviewers djmq and YYvQ (implicitly), the failure of CRL baselines to improve performance undermines the paper’s framing. The paper does not provide evidence that the "non-stationarity" here requires CRL techniques, making the benchmark ill-suited for its stated target audience.
- Evaluation Protocol: The lack of task-agnostic baselines (e.g., CLEAR) and diagnostic metrics (forgetting, forward transfer) prevents a proper assessment of algorithmic behavior.
- Theoretical Grounding: While the AC accepts the proposed experiential view, Reviewer Vnh3's objection regarding the "Stationary POMDP" nature highlights that the authors did not theoretically characterize why this specific stationary physics engine should be treated as a continual learning problem rather than a long-horizon control problem.

**Reviewer Scores:**

- Reviewer Vnh3 (2->2): Would maintain the score. The reviewer's concern that the problem framing is mismatched is empirically supported, albeit subtly: the ineffectiveness of CRL-specific mechanisms suggests that the environment's difficulty is dominated by other factors (e.g., hard exploration in a POMDP) rather than the specific continual learning challenges the authors claim to target. The benchmark thus fails to isolate the property of interest, validating the reviewer's objection to its positioning.
- Reviewer djmq (4->4): Would maintain the score. While acknowledging the new baselines, the reviewer emphasized that the evaluation protocol remains fundamentally incomplete due to the lack of task-agnostic baselines (e.g., CLEAR) and diagnostic metrics (e.g., Forgetting, Transfer). Without these, the benchmark cannot distinguish between forgetting and exploration failures.
- Reviewer CrrQ (4->4): Would maintain the score based on limited novelty. The reviewer viewed the work as a derivative "wrapper" around an existing game rather than a conceptual advance in benchmark design, arguing that high difficulty alone does not constitute scientific novelty comparable to benchmarks like ALE.
- Reviewer YYvQ (6->6): Maintained the score. The reviewer remains positive about the platform's potential utility and engineering quality, viewing the scope limitations as acceptable features rather than fatal flaws. However, the AC finds that this positive assessment focuses primarily on the software artifact and does not sufficiently weigh the critical methodological disconnect highlighted by the other three reviewers (i.e., the failure of the benchmark to validate CRL-specific claims).

---

### Decision · Program_Chairs · 2026-01-26

Reject